# Streaming Attention Approximation via Discrepancy Theory

**Ekaterina Kochetkova**
EPFL
ekaterina.kochetkova@epfl.ch

**Kshiteej Sheth**
EPFL
kshiteej.sheth@epfl.ch

**Insu Han**
KAIST
insu.han@kaist.ac.kr

**Amir Zandieh**
Google Research
zandieh@google.com

**Michael Kapralov**
EPFL
michael.kapralov@epfl.ch

## Abstract

Large language models (LLMs) have achieved impressive success, but their high memory requirements present challenges for long-context token generation. In this paper we study the streaming complexity of attention approximation, a key computational primitive underlying token generation.

Our main contribution is BalanceKV, a streaming algorithm for $\epsilon$-approximating attention computations based on geometric process for selecting a balanced collection of Key and Value tokens as per Banaszczyk's vector balancing theory. We complement our algorithm with space lower bounds for streaming attention computation. Besides strong theoretical guarantees, BalanceKV exhibits empirically validated performance improvements over existing methods, both for attention approximation and end-to-end performance on various long context benchmarks.

## 1 Introduction

Transformer-based models are the foundation of ongoing artificial intelligence revolution. Their applications span a wide range of domains, from leading-edge language models (LLM) [1, 65] to text-to-image [58, 66, 69], text-to-video synthesis [70], coding assistance [68] and even in multimodal domains across text, audio, image, and video [53]. At the core of these models is the Transformer architecture, powered by the self-attention mechanism [73], which enables effective capture of pairwise correlations across tokens in an input sequence. As these models scale in size and context length [41], they face significant computational challenges, particularly in terms of memory usage. Efficiency and accuracy are essential to unlock the full potential of LLMs in generating long sequences.

**Space bottlenecks in transformer models.** Most large language models, along with multimodal and video models, adopt an autoregressive, decoder-only architecture. This architecture generates tokens sequentially, applying attention dynamically to each newly generated token. To avoid redundant attention score computations during the generation phase, these models explicitly store the key and value embeddings of previously generated tokens in a cache in each attention layer. Thus, a major challenge is the fact that the memory complexity of storing previously generated key value embeddings scales with both the model size (i.e., the number of layers and attention heads) and, critically, the context size. Additionally, each model session typically requires its own dedicated cache for storing key value embeddings, further exacerbating memory usage. This growing demand has become a significant bottleneck, affecting both memory consumption and computational speed, particularly for models handling long context lengths.

39th Conference on Neural Information Processing Systems (NeurIPS 2025).

**Streaming attention computation.** The main reason for the need of storing the past key and value embeddings is for the attention computation happening inside each self attention layer during token generation after processing a context – to generate the next token, each self attention layer computes the attention between the query embedding of the current token and the key and value embeddings of all the tokens that were previously generated or part of the context. In this paper we study the *streaming attention approximation* problem – the problem of approximately computing attention using a small amount of space, i.e. without storing all previously seen key and value embeddings. Our main contribution is BALANCEKV, a novel provably correct algorithm for streaming attention approximation based on discrepancy theory. The core of our approach is a vector balancing algorithm from discrepancy theory that exploits the geometry of key and value tokens to deduce a small subset of them that well approximates the operations happening inside a self-attention layer. We complement our algorithm with a lower bound on the streaming complexity of approximating attention.

An algorithm for streaming attention approximation can directly be used for compressing the key value cache which stores the past key value embeddings in each layer in an LLM, thus improving the efficiency of LLM token generation. We empirically evaluate BALANCEKV both on the problem of approximating attention and on end-to-end generation tasks, showing performance gains.

## 1.1 Related Work

For discrepancy theory, Banaszczyk's seminal works [6, 7] establishing theoretical guarantees for vector set discrepancy have sparked research in the vector balancing problem [17]. This led to algorithmic developments in both offline [8] and online [9, 3, 43] settings. The vector balancing problem has particular relevance to streaming and sublinear algorithms, as minimizing a dataset's discrepancy yields small subsets that effectively preserve the original dataset's properties. Recently [55, 15] extend these discrepancy theory ideas for *kernel density estimation* using sublinear memory.

A simple yet effective approach is quantizing previously generated key value embeddings with fewer bits [80, 78, 26, 40, 49, 35, 84, 81]. Another line of work focuses on token-level pruning, where redundant or less important tokens get evicted from the set of all previously generated key value embeddings [10, 85, 48, 76, 83, 46]. Many of the works in this line have used accumulated attention scores to select important previously generated tokens [85, 46, 76]. Recent works extend those methods to an adaptive way of budget allocation across layer [14] and head [30].

## 1.2 Overview of Our Contributions

In this work we take the token subset selection approach to reduce the memory complexity of LLM token generation: store and maintain only a subset of previously generated key and value embeddings corresponding to a few "important" tokens in the sequence. Of course, the central question is how to define "importance" of tokens. Our approach here is to apply discrepancy theory, which, at a high level, considers a token important if it is crucial to preserving the projection of the total collection of tokens onto some direction in the token space. This leads to the idea of selecting a subset of tokens that is "balanced" simultaneously in every direction. Inspired by the recent breakthrough result of [3] on online discrepancy minimization, we design a method for balancing key-value pairs online using small space, namely our BALANCEKV algorithm. Interestingly, this algorithm is *online*, i.e. the importance of a token is determined only by preceding tokens – in sharp contrast with state of the art heuristics for token selection such as PyramidKV [14] and SnapKV [46], whose performance, as we show, our algorithm matches or improves upon. Our contributions are:

1. In Section 3 we propose BALANCEKV, an algorithm for recursively compressing the set of previously generated tokens using a geometric correlated sampling process based on discrepancy theory. We show that BALANCEKV gives provable guarantees for streaming attention approximation under the bounded $\ell_2$ norm assumption (Theorem 3.1). Using tools from communication complexity, we also show a lower bound on the memory complexity of any algorithm for streaming attention approximation in Section 3. Section 2 contains the formal problem formulation of streaming attention approximation, its applicability to key value cache compression, as well as a technical overview of the main results and techniques of Section 3.

2. In Section 4 we empirically evaluate our algorithm in various settings. In Section 4.1 we show our approach leads to a lower relative error for single layer attention approximation for open-source LLMs including `Llama-3.1-8B-Instruct` [27] and `Ministral-8B-Instruct-`

2410 [52] as compared to uniformly sampling keys and values in the cache. Section 4.1 we also perform ablation studies to show how various parameters in our algorithm affect the relative error for single layer attention approximation. In Sections 4.2 and 4.3 we perform end to end experiments on various benchmarks such as LongBench [5] using models of various sizes such as `Llama-3.1-8B-Instruct`,`Qwen-2.5-14B-Instruct` and `Qwen-2.5-32B-Instruct` [77, 71], and Needle in a Haystack [39]. We show that our provable method for attention approximation when applied to key value cache compression performs better compared to previous existing token subset selection heuristics on end to end tasks. Finally in Section 4.4 we present system efficiency metrics regarding our implementation.

## 2 Technical Overview

In this section, we first set up the formal problem formulation that we tackle, followed by an overview of our techniques and our main results.

### 2.1 Streaming Attention Approximation: Formulation and Motivation

Autoregressive Transformers generate tokens one by one and each depends on the previously generated tokens. When Transformers process a sequence of tokens, the *attention mechanism* operates by computing three types of embeddings for each token at every layer: query, key and value. The query and key capture how different tokens interact, while the value is the actual content to be aggregated. Such interactions are quantified by so-called *attention scores*, obtained by applying the softmax to the inner product between the query of a given token and the keys of all others. These scores determine how much each previous token's value contributes to the final output. Once the keys and values are computed for a given token, they do not need to be recomputed when generating subsequent tokens.

Formally, suppose that we have a stream of query, key and value embeddings $(q_1, k_1, v_1), \ldots, (q_n, k_n, v_n)$, that is the $j$-th token is represented as a triplet of $(q_j, k_j, v_j)$ where $q_j, k_j, v_j \in \mathbb{R}^d$ for all $j \in [n]$. Let $K_j, V_j \in \mathbb{R}^{j \times d}$ be matrices defined by stacking those keys and values in their respective rows. To compute the following at every step $j$ to generate $j + 1$ token, is called the *streaming attention problem*:

$$\text{Attn}(q_j, K_j, V_j) := \text{softmax}\left(\frac{K_j \cdot q_j}{\sqrt{d}}\right)^T \cdot V_j. \tag{1}$$

Keeping all of the key-value pairs in the cache is prohibitively expensive, especially for long sequences. Instead, we opt for approximate computation by sampling a few key-value pairs. Specifically, our goal is to construct an algorithm that at every time step $j$ computes an estimator $z_j$ for $\text{Attn}(q_j, K_j, V_j)$ in sublinear in $n$ time and memory. In particular for given precision $\varepsilon > 0$, $z_j$ should satisfy the following error constraint:

$$\|z_j - \text{Attn}(q_j, K_j, V_j)\|_2 \leq \varepsilon \left\|\text{softmax}\left(\frac{K_j \cdot q_j}{\sqrt{d}}\right)\right\|_2 \|V_j\|_F. \tag{2}$$

A sublinear in $n$ time and memory algorithm to compute $z_j$ will require knowledge of significantly less key-value pairs than $K_j, V_j$, thus reducing the size of the key value cache needed to store them. This motivates the study of streaming attention approximation, as an algorithm for this can directly be used for key value cache compression during LLM token generation. In the next section we discuss how we will construct such an estimator $z_j$ at a high level.

### 2.2 SOFTMAXBALANCE: Attention Approximation via Discrepancy Theory

We now start with presenting the main ideas of our approach. By the definition of softmax, Equation (1) can be written as

$$\text{Attn}(q_j, K_j, V_j) = \frac{1}{Z_j} \exp\left(\frac{K_j \cdot q_j}{\sqrt{d}}\right)^T \cdot V_j,$$

where for a matrix $A$ we write $\exp(A)$ to denote entry-wise exponential function to $A$ and $Z_j := \sum_{i \in [j]} \exp(\langle k_i, q_j \rangle / \sqrt{d})$. Our approach to approximate $\text{Attn}(q_j, K_j, V_j)$ consists of two subroutines which approximate:

1. Softmax normalization $Z_j = \sum_{i \in [j]} \exp(\langle k_i, q_j \rangle / \sqrt{d})$,

2. Matrix-vector product between $V_j$ and $\exp(K_j \cdot q_j / \sqrt{d})$.

To understand our main idea, suppose we are at the end of the stream (i.e., $j = n$) and we store all key-value pairs $(k_1, v_1), \ldots, (k_n, v_n)$. Then for an arbitrary query $q_n$ we aim to approximate the matrix-vector product $\exp(K_n \cdot q_n / \sqrt{d})^T \cdot V_n = \sum_{i \in [n]} \exp(\langle k_i, q_n \rangle / \sqrt{d}) v_i$ by choosing a subset of the rows of $K_n$ and $V_n$ of size at most $n/2$ which corresponds to a compression rate of 0.5. Suppose we can design an algorithm which splits the set $C$ of all keys and values into two groups $C'$ and $C \backslash C'$ so that the matrix-vector product function for any query vector $q_n$ is roughly equal over $C'$ and $C \backslash C'$ that is informally,

$$\sum_{\{k,v\} \in C'} \exp \left( \frac{\langle k, q_n \rangle}{\sqrt{d}} \right) v \approx \sum_{\{k,v\} \in C \backslash C'} \exp \left( \frac{\langle k, q_n \rangle}{\sqrt{d}} \right) v.$$

Then, we are able to approximate the matrix-vector product function with either one of the sums above since informally:

$$\sum_{\{k,v\} \in C} \exp \left( \frac{\langle k, q_n \rangle}{\sqrt{d}} \right) v \approx 2 \sum_{\{k,v\} \in C'} \exp \left( \frac{\langle k, q_n \rangle}{\sqrt{d}} \right) v.$$

Therefore, it would suffice to keep the smaller subset of $C'$ and $C \backslash C'$ as the desired subset of key value embeddings and discard the rest. If we wanted to compress the key value cache to a smaller size by a factor $2^T$ for some $T$, we would recursively compress the selected subset using the same procedure $T - 1$ more times.

A similar goal is captured by the *vector balancing problem* studied extensively in discrepancy theory; given a set of vectors $C = \{k_1, \ldots, k_n\} \subset \mathbb{R}^d$ with $\|k_j\|_2 \leq 1$ for all $j$, partition them into two groups $C', C \setminus C'$ such that for any $q \in \mathbb{R}^d$ it holds $\sum_{k \in C'} \langle k, q \rangle \approx \sum_{k \in C \setminus C'} \langle k, q \rangle$ with high probability. The Self-Balancing Walk algorithm [3] is a breakthrough result for the above vector balancing problem. However we need to develop an algorithm for the vector balancing problem with respect to function $\exp(\langle k, \cdot \rangle / \sqrt{d}) v$ instead of the inner product function $\langle k, \cdot \rangle$.

Our first contribution is to develop an algorithm for our task, building upon the result from the self-balancing walk [3], which essentially randomly partitions the set of keys and values $C$ into $C'$ and $C \setminus C'$ such that the following holds with high probability under the assumptions that the norms of the query and key embeddings are bounded,

$$\left\| \sum_{\{k,v\} \in C'} \exp \left( \frac{\langle k, q_n \rangle}{\sqrt{d}} \right) v - \sum_{\{k,v\} \notin C'} \exp \left( \frac{\langle k, q_n \rangle}{\sqrt{d}} \right) v \right\|_2 \leq O\left( \log(nd) \right) \cdot \max_{j \in [n]} \|v_i\|_2.$$

We refer to this algorithm as SOFTMAXBALANCE, its formal guarantee is presented in Theorem 3.3 and its pseudocode is presented in Algorithm 2. Theorem 3.3 shows that SOFTMAXBALANCE succeeds to divide $C$ into subsets $C'$ and $C \backslash C'$ which are balanced with respect to function $\exp(\langle k, \cdot \rangle / \sqrt{d}) v$ up to an error which only has logarithmic dependence on the size of $C$. In addition, SOFTMAXBALANCE can accept as input value vectors of arbitrary dimension $s$. Therefore, if instead of the value vectors $v_1, \ldots, v_n \in \mathbb{R}^d$ we input the set of scalars $v_1 = \cdots = v_n = 1$, we will get an algorithm for the vector balancing problem with respect to function $\exp(\langle k, \cdot \rangle / \sqrt{d})$. This implies that we can use SOFTMAXBALANCE to compress the key value cache to even approximate the softmax normalization $\sum_{i \in [n]} \exp(\langle k_i, q_n \rangle / \sqrt{d})$. We now discuss how to use SOFTMAXBALANCE for streaming attention approximation, i.e. to use it to compute an estimator $z_j$ satisfying Equation (2).

### 2.3 BALANCEKV: Implementing SOFTMAXBALANCE in Streaming

For a sequence of $n$ tokens and a given memory budget of $t \ll n$, we aim to design a procedure which applies SOFTMAXBALANCE to select from $n$ key-value embeddings a set of at most $t$ in the streaming setting and can compute an estimator $z_j$ satisfying Equation (2) for all steps $j$ in the stream. In the streaming setting one needs to consider the following aspects. As described in the previous section, one iteration of SOFTMAXBALANCE only allows one to select a $n/2$ sized subset of

$n$ key-value embeddings, which is higher than the desired budget of $t$ embeddings. This can be easily mitigated by recursively applying SOFTMAXBALANCE $2^{\log(n/t)}$ times, each time halving the set of key-value embeddings. However, this cannot be implemented in the streaming as we have a limited memory budget of $t$ which prohibits us from storing all key-value embeddings during recursion.

To deal with this, we use the classical merge and reduce technique used in the design of streaming algorithms [13, 51, 33]. MERGEANDREDUCE algorithm is a recursive binary tree-based approach that allows one to implement SOFTMAXBALANCE recursively in a streaming setting with the total memory not exceeding $\widetilde{O}(dt)$, where $\widetilde{O}(\cdot)$ supresses polynomial in $\log n$ factors, under the assumption that the norms of queries and keys are bounded. The guarantees of MERGEANDREDUCE are presented in Theorem 3.4, its pseudocode in Algorithm 4 and a visual representation in Figure 2. If the norms of all value embeddings in the stream are the same up to constant factors, that is for all $i, j \in [n]$ $0.5 \leq \|v_i\|_2 / \|v_j\|_2 \leq 2$, then the outputs of MERGEANDREDUCE can be used to construct an estimator $z_j$ satisfying our attention approximation guarantee of equation Equation (2) with precision $\varepsilon$ for $t = \widetilde{O}(\sqrt{d}/\varepsilon)$. However, the value embeddings may have very different norms.

Our main algorithm BALANCEKV (pseudocode in Algorithm 1) deals with this issue by grouping the key-value embeddings in the stream according to the norms of the value embeddings, running a separate instance of MERGEANDREDUCE on each group, and combining the outputs of each instance of MERGEANDREDUCE. BALANCEKV constructs a final estimator $z_j$ satisfying Equation (2) with precision $\varepsilon$ only using $\widetilde{O}(d\sqrt{d}/\varepsilon)$ memory and $\widetilde{O}(d^2/\varepsilon^2)$ runtime per every step $j$ of the stream, assuming the norms of query and key embeddings are bounded. Existing methods [83] subsample keys and values independently in the cache, and thus have a $1/\varepsilon^2$ dependence on $\varepsilon$ in total memory. The guarantees of BALANCEKV are presented in Theorem 3.1.

Finally using the lower bound on the communication complexity of INDEX, we show a lower bound on the memory complexity of any algorithm for streaming attention approximation in Theorem 3.2.

## 3 Main Theoretical Results

Our main algorithm for streaming attention approximation is BALANCEKV. It takes in as input a stream of $n$ tokens $(q_1, k_1, v_1), (q_2, k_2, v_2), \ldots, (q_n, k_n, v_n)$ and at every step of the stream outputs an estimate $z_j$ to $\text{Attn}(q_j, K_j, V_j)$ (see Equation (1) for the definition of Attn(.)) satisfying Equation (2) with precision $\varepsilon$. Assuming that the $\ell_2$ norms of $q_j, k_j$ are at most $r$ for all $j$, BALANCEKV uses total space $\widetilde{O}(d\sqrt{d}e^{2r^2/\sqrt{d}} \cdot 1/\varepsilon)$ and uses $\widetilde{O}(d^2 e^{4r^2/\sqrt{d}} \cdot 1/\varepsilon^2)$ runtime at each step $j$ of the stream to output $z_j$. Our main theorem is as follows.

**Theorem 3.1.** *For any $r, \varepsilon > 0$, any positive integers $n, d$, any set of tokens $(q_1, k_1, v_1), (q_2, k_2, v_2), \ldots, (q_n, k_n, v_n)$ where $q_j, k_j, v_j \in \mathbb{R}^d$ satisfy $\|q_j\|_2, \|k_j\|_2 \leq r$ for all $j$, consider an invocation of BALANCEKV with*

$$\text{batch size } t = \widetilde{O}\left(\sqrt{d}e^{2r^2/\sqrt{d}}/\varepsilon\right) \text{ and compression rate } 2^{-T} \text{ with } T = \log(n/t).$$

*Then BALANCEKV outputs a vector $z_j$ satisfying Equation (2) with probability at least $1 - 1/poly(n)$ at every step $j$ of the stream. It uses total memory $\widetilde{O}\left(d\sqrt{d}e^{2r^2/\sqrt{d}}/\varepsilon\right)$ across all steps of the stream and runtime of $\widetilde{O}\left(d^2 e^{4r^2/\sqrt{d}}/\varepsilon^2\right)$ per step of the stream.*

A pseudocode of BALANCEKV is described in Algorithm 1. At its core BALANCEKV relies on our main discrepancy based algorithm, namely SOFTMAXBALANCE– see Section 3.1 for details on SOFTMAXBALANCE. BALANCEKV uses the output of SOFTMAXBALANCE to compute estimates of the numerator and denominator of $\text{Attn}(q_j, K_j, V_j)$ and returns the desired attention approximation $z_j$ for each streamed index $j$. There are two subtleties, however. First, it is important to bucket tokens in the stream according to the norm of the value vectors – see lines 5 and 6. Second, a direct application of SOFTMAXBALANCE would require too much memory space. To ensure small space usage, we apply a classical streaming technique, namely the MERGEANDREDUCE algorithm on top of SOFTMAXBALANCE to reduce the space consumption. The space reduction achieved by MERGEANDREDUCE is by running a logarithmic number of copies of SOFTMAXBALANCE in a tree-like fashion. More details are introduced in Section 3.2.

---

**Algorithm 1** BALANCEKV$((q_j, k_j, v_j)_{j=1}^n, r, t, T, \varepsilon)$

---

1: **input:** stream of $n$ tokens $(q_j, k_j, v_j)$, diameter $r$, batch size $t$, compression rate $2^{-T}$, precision parameter $\varepsilon$.
2: `// Bucket the stream and maintain` $\log(n)$ `instances of MERGEANDREDUCE,`
   `MR-NUMERATOR`$_i$`, for each bucket to approximate the numerator of Attn`$(q_j, K_j, V_j)$`;`
   `and one instance, MR-DENOMINATOR, to approximate its denominator.`
3: $v_{\max} \leftarrow 0$
4: **repeat**
5:    Find an index $i$ such that $2^i \geq \|v_j\|_2 \geq 2^{i-1}$
6:    Send $(k_j, v_j)$ as input to MR-NUMERATOR$_i$             `// Bucket the stream by` $\|v\|_2$
7:    $v_{\max} \leftarrow \max\{\|v_j\|_2, v_{\max}\}$
8:    Erase all MR-NUMERATOR$_i$ with $2^i \leq \frac{\varepsilon}{2n} e^{-\frac{r^2}{\sqrt{d}}} v_{\max}$      `// Erase small norm buckets`
9:    $C_i^0, \ldots, C_i^T \leftarrow$ the output of MR-NUMERATOR$_i$
10:    $V^l \leftarrow \cup_i C_i^l$ for $l = 0, \ldots, T$          `// Combine the outputs of MR-NUMERATOR`$_i$
11:    Send $(k_j, 1)$ as input to MR-DENOMINATOR
12:    $K^0, \ldots K^T \leftarrow$ MR-DENOMINATOR
13:    **output:** $z_j = \dfrac{\sum_{l=0}^T 2^l \sum_{\{k,v\} \in V^l} \exp\left(\frac{\langle k, q_j \rangle}{\sqrt{d}}\right) v}{\sum_{l=0}^T 2^l \sum_{\{k,v\} \in K^l} \exp\left(\frac{\langle k, q_j \rangle}{\sqrt{d}}\right)}$
14:    $j \leftarrow j + 1$
15: **until** token stream ends

---

To summarize, BALANCEKV groups tokens in the stream according to the norms of the corresponding value embeddings, runs a separate instance of MERGEANDREDUCE on each group, and combines the outputs of each instance to construct the final estimate for Attn$(q_j, K_j, V_j)$ at each step $j \in [n]$. Next we present SOFTMAXBALANCE and MERGEANDREDUCE. The full proof of Theorem 3.1 is given in appendix Section A.1. Finally we state the theorem which provides a lower bound on the memory complexity of any algorithm for streaming attention approximation below, its full proof is provided in appendix Section C.

**Theorem 3.2.** *Suppose that* $r^2 \leq d$. *Any streaming algorithm which on input* $(\{k_1, v_1\}, \ldots, \{k_n, v_n\}, q)$, $\|q\|_2, \|k_i\|_2 \leq r$, *outputs* $z_q$ *satisfying Equation* (2) *with probability 0.999 has space complexity* $\Omega\left(\min\{\frac{1}{\varepsilon^2}, d \exp(2r^2/\sqrt{d})\}\right)$.

### 3.1 SOFTMAXBALANCE

We now present our main discrepancy based compression algorithm, SOFTMAXBALANCE. Given a sequence of key and value embeddings $C = \{(k_1, v_1), \ldots (k_n, v_n)\}$ (with key and value embeddings having possibly different dimensions), the goal of SOFTMAXBALANCE is to produce a partition of $C$ into subsets $C', C \setminus C'$ such that for any query $q \in \mathbb{R}^d$ we have that $\sum_{(k,v) \in C'} \exp(\langle k, q \rangle / \sqrt{d}) v \approx \sum_{(k,v) \in C \setminus C'} \exp(\langle k, q \rangle / \sqrt{d}) v$ with high probability. Without loss of generality assume that $|C'| \leq |C|/2$, we can then output $2 \sum_{(k,v) \in C'} \exp(\langle k, q \rangle / \sqrt{d}) v$ as an approximation to $\sum_{(k,v) \in C} \exp(\langle k, q \rangle / \sqrt{d}) v$, thus achieving a factor 2 compression. Its description is presented in Algorithm 2 below. We note that while SOFTMAXBALANCE takes as input a sequence of key and value embeddings, it can nevertheless be used to compute the softmax normalization: we simply run it on the keys, with the corresponding value vector one-dimensional and all equal to 1 – see line 11 in BALANCEKV, where SOFTMAXBALANCE is called within the corresponding invocation of MERGEANDREDUCE with value vectors as 1s. It's guarantees are as follows.

**Theorem 3.3.** *Given sets* $K = \{k_1, \ldots, k_n\} \subset \mathbb{R}^d, V = \{v_1, \ldots, v_n\} \subset \mathbb{R}^s$, *and failure probability* $\delta > 0$, *define* $C$ *to be the dataset of pairs* $C = \{(k_1, v_1), \ldots, (k_n, v_n)\}$. *There exists a randomized algorithm,* SOFTMAXBALANCE, *which outputs a subset* $C' \subset C, |C'| \leq |C|/2$, *such that, for any*

---

**Algorithm 2** SOFTMAXBALANCE$((k_j, v_j)_j, r_{\text{key}}, r_{\text{value}}, \delta)$

---

1: **input:** stream of $\leq n$ key-value embeddings $(k_j, v_j)$, radii $r_{\text{key}}, r_{\text{value}}$: $\max_j \|k_j\|_2 \leq r_{\text{key}}$, $\max_j \|v_j\|_2 \leq r_{\text{value}}$, probability of failure $\delta$.
2: $R \leftarrow \exp(r_{\text{key}}^2 / 2\sqrt{d}) \cdot r_{\text{value}}$
3: $c \leftarrow 30 \log(n/\delta)$
4: Initialize zero vector $\eta \leftarrow \{0\}$
5: **for** $j$ from 1 and until the end of the stream **do**
6: $\quad y \leftarrow \left( \exp(\langle k_i, k_j \rangle / \sqrt{d}) \langle v_i, v_j \rangle \right)_{i \in [j]}$
7: $\quad$ **if** $\left| y^T \eta \right| > c \cdot R^2$ **then** FAIL
8: $\quad p_j \leftarrow \frac{1}{2} - \frac{y^T \eta}{2c \cdot R^2}$
9: $\quad \eta_j \leftarrow \begin{cases} +1 & \text{with probability } p_j \\ -1 & \text{o.w.} \end{cases}$
10: $\quad$ Add a new zero coordinate $\eta_{j+1} \leftarrow 0$
11: **end for**
12: **if** $|\{(k_i, v_i) : \eta_i = 1\}| \leq |\{(k_i, v_i) : \eta_i = -1\}|$ **then**
13: $\quad$ **output:** $\{(k_i, v_i) : \eta_i = 1\}$
14: **else**
15: $\quad$ **output:** $\{(k_i, v_i) : \eta_i = -1\}$
16: **end if**

---

*vector $q \in \mathbb{R}^d$, with probability at least $1 - \delta$,*

$$\left\| \sum_{\{k,v\} \in C'} \exp\left( \frac{\langle k, q \rangle}{\sqrt{d}} \right) v - \sum_{\{k,v\} \notin C'} \exp\left( \frac{\langle k, q \rangle}{\sqrt{d}} \right) v \right\|_2 \leq$$

$$O\left( \sqrt{s} \cdot \log(ns/\delta) \cdot \exp\left( \frac{\|q\|_2^2}{2\sqrt{d}} \right) \cdot \exp\left( \max_{j \in [n]} \frac{\|k_j\|_2^2}{2\sqrt{d}} \right) \cdot \max_{j \in [n]} \|v_j\|_2 \right).$$

*The runtime of SOFTMAXBALANCE is $O((d + s)n^2)$ and memory is $O((d + s)n)$.*

The proof of the above theorem uses the breakthrough result of [3] for the vector balancing problem, one of the main problems in discrepancy theory. Given a set of vectors $k_1, \ldots, k_n$ the result of [3] produces a subset $C$ of these vectors of at most half the size such that for any vector $q$ we have that $\sum_{k \in C} \langle k, q \rangle \approx \sum_{k \in [n] \setminus C} \langle k, q \rangle$ with high probability. Our main contribution is an algorithm for the vector balancing problem with respect to the function $\exp(\langle k, \cdot \rangle / \sqrt{d}) v$ as compared to $\langle k, \cdot \rangle$ in the case of [3]. We defer the proof of Theorem 3.3 to Appendix A.2.

### 3.2 MERGEANDREDUCE

As briefly mentioned above in Section 3, MERGEANDREDUCE is a streaming version of SOFT-MAXBALANCE. The idea is to partition the stream of tokens into batches of size $t$, apply SOFT-MAXBALANCE to the batches to reduce the size of each batch by a constant factor, and then repeat recursively – see Fig. 2 in the appendix.

If we set batch size $t$ to be about $1/\varepsilon$ (see Theorem 3.4 below for the more precise setting), we obtain a streaming algorithm that approximates $\sum_{i=1}^j \exp(\langle k_i, q_j \rangle / \sqrt{d}) v_i$ at any point $j$ in the stream using total space $\widetilde{O}(d\sqrt{d}e^{2r^2/\sqrt{d}}/\varepsilon)$ and runtime $\widetilde{O}(d^2 e^{4r^2/\sqrt{d}}/\varepsilon^2)$ per step, where $r$ is an upper bound on the norms of key and query embeddings.

As before, an important aspect is that MERGEANDREDUCE can handle value embeddings of dimension not necessarily equal to that of key and query embeddings. Thus, when run on scalars $v_i = 1$ for all $i$, it can also be used to approximate softmax normalization at any point $j$ in the stream. This is the main subroutine used in BALANCEKV to approximate $\text{Attn}(q_j, K_j, V_j)$. Its pseudocode description is presented in Appendix A.3.1, and its proof is in Appendix A.3.2. The formal guarantees are

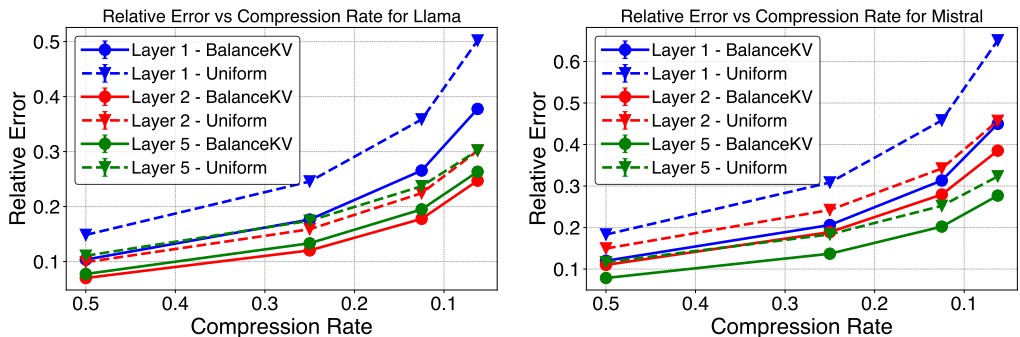

Figure 1: Comparison of relative errors across different layers of `Llama-3.1-8B-Instruct` (left) and `Ministral-8B-Instruct-2410` (right) on TriviaQA dataset.

**Theorem 3.4.** *For any $r, \varepsilon > 0$, any set of tokens $(q_1, k_1, v_1), \ldots, (q_n, k_n, v_n)$ where $q_j, k_j \in \mathbb{R}^d$ satisfy $\|q_j\|_2, \|k_j\|_2 \le r$, $v_j \in \mathbb{R}^s$ for $s \le d$ suppose,*

$$\text{batch size } t = \widetilde{O}(\sqrt{s} e^{2r^2/\sqrt{d}}/\varepsilon) \text{ and compression rate } 2^{-T} \text{ with } T = \log(n/t).$$

*Then* MERGEANDREDUCE *on input parameters $t, r, d, s, \varepsilon$, outputs at every step $j$ of the stream subsets of key-value embedding pairs $C^0, \ldots, C^T \subset C := \{(k_1, v_1), \ldots, (k_n, v_n)\}$ such that, $z_j := \sum_{i=0}^{T} 2^i \sum_{\{k,v\} \in C^i} \exp\left(\frac{\langle k, q_j \rangle}{\sqrt{d}}\right) v$, satisfies with probability at least $1 - 1/poly(n)$,*

$$\left\| \sum_{i=1}^{j} \exp\left(\frac{\langle k_i, q_j \rangle}{\sqrt{d}}\right) v_i - z_j \right\|_2 \le \varepsilon j \cdot e^{-r^2/\sqrt{d}} \cdot \max_{i \in [n]} \|v_i\|_2.$$

*Total memory of the algorithm is $\widetilde{O}(d\sqrt{s} e^{2r^2/\sqrt{d}}/\varepsilon)$, its $j$-th iteration runtime is $\widetilde{O}(dse^{4r^2/\sqrt{d}}/\varepsilon^2)$.*

# 4 Experiments

In this section we now present our experimental results. The full details of all sections as well as the experimental setup and implementation can be found in Appendix B.

## 4.1 Ablation Studies on Single Layer Attention Approximation

We evaluate the effectiveness of BALANCEKV for approximating attention in individual layers of `Llama-3.1-8B-Instruct` [27] and `Ministral-8B-Instruct-2410` [52] on the TriviaQA dataset from LongBench [5]. Specifically, we examine layers 1, 2, and 5 and compare against independent uniform sampling key and value embeddings.

Due to space limitations, we provide the full experimental details in Appendix B.1. For each layer, we approximate attention for recent tokens using a compressed cache that retains a fixed number of initial and recent embeddings, alongside intermediate ones selected via BALANCEKV, and measure its relative error against exact attention. We vary the compression rate $2^{-T} \in \{1/2, 1/4, 1/8, 1/16\}$. As shown in Fig. 1, BALANCEKV consistently yields lower relative approximation error than approximating attention by uniform sampling past key value pairs across all settings, empirically validating its advantage as predicted by Theorem 3.1.

For a fixed dataset and layer, we also analyzed how the performance and runtime of BALANCEKV depend on the batch size and compression rate. More precisely, we repeat the single-layer attention approximation experiment TriviaQA and layers 1 and 15 of `Llama-3.1-8B-Instruct`, for batch size $\in [64, 128, 256]$ and compression rate $2^{-T} \in [1/2, 1/4, 1/8]$. The results are presented in Figure 3. As this experiment suggests, the quality of attention approximation increases as the size of the block doubles, while the runtime becomes slower as also proven theoretically.

| Method | qasper | multi | hotpotqa | 2wiki | gov | multinews | trec | triviaqa | samsum | p.count | p.ret | lcc | repo-p | average |
|---|---|---|---|---|---|---|---|---|---|---|---|---|---|---|
| **Qwen2.5-32B-Instruct** | | | | | | | | | | | | | | |
| Exact (Baseline) | 44.56 | 50.65 | 69.14 | 60.39 | 21.3 | 19.52 | 75.33 | 81.14 | 43.17 | 22.0 | 99.67 | 50.9 | 35.22 | 51.77 |
| StreamingLLM | 20.12 | 34.35 | 51.84 | 48.23 | 19.09 | 17.10 | 61.00 | 51.14 | 28.52 | 23.33 | 41.33 | 39.19 | 26.54 | 35.52 |
| PyramidKV | 34.47 | 46.33 | 67.78 | 55.92 | 15.16 | 15.39 | 69.33 | 63.24 | 40.36 | 22.67 | 99.33 | 48.32 | 34.35 | 47.13 |
| SnapKV | 36.21 | 46.78 | 66.64 | 57.02 | 16.35 | 16.07 | 70.33 | 77.53 | 41.08 | 22.0 | 99.33 | 49.04 | 35.62 | 48.77 |
| Uniform | 39.28 | 43.82 | 64.82 | 57.84 | 23.10 | 19.50 | 73.00 | 81.63 | 39.70 | 22.00 | 92.00 | 44.97 | 32.19 | 48.76 |
| BALANCEKV | 40.14 | 43.17 | 64.46 | 58.06 | 22.26 | 20.32 | 73.00 | 80.68 | 41.07 | 22.33 | 92.0 | 44.95 | 32.43 | **48.84** |
| **Qwen2.5-14B-Instruct** | | | | | | | | | | | | | | |
| Exact (Baseline) | 43.39 | 52.63 | 64.06 | 53.71 | 28.07 | 22.4 | 74.67 | 88.75 | 44.81 | 22.33 | 99.0 | 63.69 | 46.3 | 54.14 |
| StreamingLLM | 20.73 | 32.62 | 49.93 | 42.39 | 21.63 | 18.69 | 59.67 | 74.96 | 29.57 | 11.67 | 63.0 | 46.16 | 32.25 | 38.71 |
| PyramidKV | 31.76 | 46.6 | 62.83 | 50.0 | 19.08 | 17.68 | 65.0 | 85.52 | 42.61 | 22.0 | 99.33 | 60.62 | 44.4 | 49.8 |
| SnapKV | 32.95 | 47.53 | 61.96 | 50.34 | 20.29 | 18.28 | 60.67 | 88.75 | 42.97 | 22.67 | 99.33 | 61.32 | 45.84 | 50.22 |
| Uniform | 37.07 | 41.69 | 61.11 | 49.96 | 29.18 | 22.67 | 71.67 | 87.89 | 40.34 | 22.0 | 84.33 | 58.0 | 42.57 | 49.88 |
| BALANCEKV | 37.02 | 41.96 | 61.74 | 50.9 | 29.26 | 22.64 | 71.67 | 88.1 | 41.14 | 23.67 | 87.67 | 58.64 | 43.63 | **50.62** |
| **Llama-3.1-8B-Instruct** | | | | | | | | | | | | | | |
| Exact (Baseline) | 42.87 | 48.54 | 52.05 | 38.6 | 31.31 | 22.07 | 71.67 | 91.85 | 42.36 | 20.37 | 98.13 | 49.62 | 42.73 | 50.17 |
| StreamingLLM | 20.65 | 30.71 | 39.14 | 32.43 | 23.10 | 18.70 | 58.00 | 83.87 | 28.85 | 20.36 | 97.26 | 33.69 | 30.46 | 39.79 |
| PyramidKV | 33.86 | 39.75 | 47.12 | 35.96 | 20.03 | 17.78 | 63.67 | 90.76 | 40.21 | 20.4 | 98.97 | 45.25 | 39.51 | 45.64 |
| SnapKV | 33.91 | 42.55 | 49.09 | 36.13 | 20.48 | 17.67 | 62.0 | 91.7 | 40.23 | 20.33 | 98.86 | 46.7 | 39.86 | 46.12 |
| Uniform | 37.18 | 37.15 | 47.49 | 37.82 | 27.56 | 21.06 | 68.67 | 90.48 | 36.13 | 20.33 | 96.26 | 45.72 | 37.09 | 46.38 |
| BALANCEKV | 35.75 | 37.04 | 46.37 | 36.24 | 27.09 | 20.84 | 69.0 | 90.88 | 37.88 | 20.39 | 96.65 | 48.45 | 41.4 | **46.77** |

Table 1: Comparison of various cache compression methods on LongBench-E using various models. The best results among compression methods for each model are highlighted in bold.

## 4.2 End-to-end Evalution on LongBench

Next we evaluate BALANCEKV on LongBench dataset [5], which tests long-context understanding across tasks like QA, summarization, few-shot learning, synthetic reasoning, and code completion. Specifically, we test a version of uniform length distribution (LongBench-E). During inference, we compress the key-value cache in the prefill stage using a uniform compression rate of (approximately) 0.25 across all methods, while retaining all streamed embeddings during the decoding phase. We compare against StreamingLLM [76], SnapKV [46], PyramidKV [14], and uniform sampling (see Section 4.1), using their implementations from MInference [37]. The evaluation follows the Long-Bench protocol, using three pre-trained models at different scales including Llama-3.1-8B-Instruct, Qwen-2.5-14B-Instruct and Qwen-2.5-32B-Instruct. The results are reported in Table 1.

Notably, BALANCEKV consistently achieves the best overall performance among compression methods and across all models, demonstrating its effectiveness in preserving model quality for cache compression. The full experimental setup details can be found in Appendix B.2.

## 4.3 Needle-In-A-Haystack Benchmark

We evaluate BALANCEKV on the "Needle-In-A-Haystack" benchmark [39], comparing it against SnapKV, PyramidKV, StreamingLLM, and uniform sampling using Llama-3.1-8B-Instruct. The test challenges the model to retrieve a specific sentence (the "needle") embedded at an arbitrary position within a long context (the "haystack"). Following the setup in [29], we hide the needle at varying depths, from 0% to 100% of the total context length, across documents ranging from approximately 4K to 100K tokens. As in the previous experiments, all methods are evaluated under a fixed compression ratio of approximately 0.25.

To further enhance performance, we introduce an augmented version of BALANCEKV that deterministically preserves a small set of tokens whose key embeddings are strongly anti-correlated with the rest. The standard BALANCEKV procedure is then applied to the remaining tokens within each layer. As a result, BALANCEKV achieves an average accuracy of 0.99, outperforming SnapKV (0.83), PyramidKV (0.90), StreamingLLM (0.31), and uniform sampling (0.90). Detailed heatmaps of performance across different context lengths and needle depths are in Figure 4 in Appendix B.3.

## 4.4 System Efficiency Metrics

We measure wall-clock times for both the prefill stage (including cache compression) and the decoding stage using a random input of length 16,384 tokens, followed by the generation of 1,024 tokens.

Results are averaged over 10 independent runs, with the minimum runtime reported to enhance robustness. The full results are provided in Table 2 in Appendix B.4.

All compression methods incur some prefill overhead compared to the uncompressed baseline (Exact). While StreamingLLM achieves the fastest decoding speed, it suffers from significantly lower accuracy (see Table 1). Among the remaining methods, BALANCEKV achieves the lowest prefill latency and consistently delivers the best trade-off between efficiency and accuracy. This demonstrates that our discrepancy-based approach not only scales well in theory but also brings practical gains in end-to-end system performance, making it a compelling choice for real-world deployment scenarios.

### 4.5 Additional Experiments

We additionally conduct the following experiments and provide their results in Appendix B.5 due to the space limitation.

1. Our main theorem (Theorem 3.4) relies on an upper bound of $\ell_2$ norms of both query and key vectors. To validate this, we investigate the $\ell_2$ norms of queries, keys, and values (QKV) on the TriviaQA dataset from LongBench [5] using `Llama-3.1-8B-Instruct`. Specifically, we analyze prompts in TriviaQA and compute the average $\ell_2$ norms of all QKV vectors across all layers and attention heads during the prefill stage. The key findings are that all QKV norms consistently concentrate around some constants (15 for query, 15 for key, and 3 for value) with small confidence intervals (CI). Importantly, the norms remain stable across a wide range of sequence lengths, suggesting that these norms do not grow with input sequence length.

2. We perform the evaluation of BALANCEKV and the uniform sampling when applied to the `InternVL2.5-8B` multimodal LLM for compression rates 1/4 and 1/16, for evaluation on the MS COCO image captioning dataset. The experiment was run on a NVIDIA A100 GPU with 80 GB VRAM.

3. We repeat the experiment in Section 4.2 in the extremely low compression rate regime on some of the datasets from LongBench [5]. More specifically, we compress the key-value cache in the prefill stage of inference using a uniform compression rate of (approximately) 0.8, 0.9, and 0.95 with uniform sampling as well as BALANCEKV, while retaining all streamed embeddings during the decoding phase. BALANCEKV demonstrates improved performance over uniform sampling across each of the compression rates and datasets.

4. We augment Section 4.2 by adding comparison to ClusterGen [83] using `Llama-3.1-8B-Instruct`. The results are reported in Table 6.

## 5 Conclusion

We propose BALANCEKV, a token pruning method grounded in discrepancy theory. BALANCEKV enables approximate attention computation, which we both establish theoretically and validate empirically. To demonstrate the effectiveness of BALANCEKV as a KV cache compression algorithm, we conduct end-to-end experiments on a range of popular benchmarks and models of varying sizes. Finally, our work introduces a theoretical problem of optimal streaming attention space complexity.

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

# A  Full Proofs

## A.1  Proof of Theorem 3.1

Finally equipped with Theorem 3.3 and Theorem 3.4 we now state the proof of the main Theorem 3.1.

*Proof of Theorem 3.1.*  Recall that BALANCEKV approximates attention by finding good estimations for the numerator and the denominator of attention separately. In line (8), we erase those terms from the numerator, whose value vectors have sufficiently small $\ell_2$ norms. In what follows, we:

- Bound the space and time requirements of BALANCEKV, as well as it's probability of failure. Each of the bounds readily follows from Theorem 3.4,

- Bound the contribution of the erased terms to attention,

- Show that BALANCEKV (or, more concretely, procedures MR-NUMERATOR$_i$) approximate the rest of the terms of the numerator well,

- Show that BALANCEKV (its subroutine MR-DENOMINATOR) approximates the denominator of attention well.

We begin by analyzing the time/space requirements and probability of success of BALANCEKV. It never runs MR-NUMERATOR$_i$ for $i > \log_2(v_{\max})$ and $i < \log_2(\varepsilon \cdot n^{-1/2} \cdot v_{\max})$, so it never keeps more than $\log_2(\sqrt{n}/\varepsilon) = O(\log(n))$ of them. BALANCEKV at any step $j$ performs one iteration of procedure MR-DENOMINATOR, one iteration of MR-NUMERATOR$_i$ with $2^i \geq \|v_j\|_2 \geq 2^{i-1}$, computes the subsets $K_1, \ldots, K_T, V_1, \ldots, V_T$ and computes a function of the selected points in the subsets in line. Therefore, the runtime of BALANCEKV during one iteration is bounded by the maximum of the runtime of MERGEANDREDUCE during one iteration and time to compute the output. The latter is equal to its total memory. This maximum, by definition of $t$, is equal to $\widetilde{O}(d^2 e^{4r^2/\sqrt{d}}/\varepsilon^2)$. The memory of the algorithm is the union of memory of MR-NUMERATOR$_i$ for all $i$ and MR-DENOMINATOR, so the space complexity of the algorithm is $\widetilde{O}(d\sqrt{d}e^{2r^2/\sqrt{d}}/\varepsilon)$. The failure probability is bounded by union bounding the failure probabilities of all instances of MERGEANDREDUCE and at most $n$ queries in the stream, and is equal to $1/\text{poly}(n)$.

Next, we bound the contribution of the erased terms. Formally, let $v_{\max}(j) := \max_{i \leq j} \|v_i\|_2$ and define $i(j) := \max_i \left\{ 2^i \leq \frac{\varepsilon}{\sqrt{n}} v_{\max} \right\}$. Observe that

$$
\left\| \frac{\sum_{\substack{i:i\leq j \\ \|v_i\|_2 \leq 2^{i(j)}}} \exp\left(\frac{\langle k_i, q_j \rangle}{\sqrt{d}}\right) v_i}{\sum_{i=1}^{j} \exp\left(\frac{\langle k_i, q_j \rangle}{\sqrt{d}}\right)} \right\|_2 \leq \frac{\sum_{\substack{i:i\leq j \\ \|v_i\|_2 \leq 2^{i(j)}}} \exp\left(\frac{\langle k_i, q_j \rangle}{\sqrt{d}}\right) \|v_i\|_2}{\sum_{i=1}^{j} \exp\left(\frac{\langle k_i, q_j \rangle}{\sqrt{d}}\right)} \qquad \text{by triangle inequality}
$$

$$
\leq \frac{\varepsilon}{2\sqrt{n}} \cdot \frac{\sum_{\substack{i:i\leq j \\ \|v_i\|_2 \leq 2^{i(j)}}} \exp\left(\frac{\langle k_i, q_j \rangle}{\sqrt{d}}\right)}{\sum_{i=1}^{j} \exp\left(\frac{\langle k_i, q_j \rangle}{\sqrt{d}}\right)} \qquad \text{by the definition of } i(j)
$$

$$
\leq \frac{\varepsilon}{2\sqrt{n}} v_{\max}
$$

$$
\leq \varepsilon \cdot \left\| \text{softmax}\left(\frac{K_j \cdot q_j}{\sqrt{d}}\right) \right\|_2 \cdot \|V_j\|_F,
$$

$$
\tag{3}
$$

where the last inequality follows from the general inequality $\left\| \text{softmax}\left(\frac{K_j \cdot q_j}{\sqrt{d}}\right) \right\|_2 \geq \frac{1}{\sqrt{j}} \geq \frac{1}{\sqrt{n}}$ and $\|V_j\|_F \geq v_{\max}$.

We analyze the quality of approximation of the denominator together with the quality of approximation of the numerator, as both follow from Theorem 3.4. At time step $j$, procedure MR-DENOMINATOR

returns subsets $K_1, \ldots, K_T$ such that

$$\left| \sum_{i \in [j]} \exp\left(\frac{\langle k_i, q_j \rangle}{\sqrt{d}}\right) - \sum_{l=0}^{T} 2^l \sum_{\{k,v\} \in K^l} \exp\left(\frac{\langle k, q_j \rangle}{\sqrt{d}}\right) \right| \leq \varepsilon \cdot j \cdot e^{\frac{-r^2}{\sqrt{d}}} \leq \varepsilon \cdot \sum_{i \in [j]} \exp\left(\frac{\langle k_i, q_j \rangle}{\sqrt{d}}\right) \tag{4}$$

as follows from Theorem 3.4 by plugging in scalars $v_1 = \ldots = v_j = 1$. Next, define $P_{i,j} = \{\{k_l, v_l\} : l \leq j, 2^i \geq \|v_l\|_2 \geq 2^{i-1}\}$. Intuitively, $P_{i,j}$ aggregates all of the tokens proccessed by MR-NUMERATOR$_i$ which appeated before time step $j$. MR-NUMERATOR$_i$ returns subsets $V_i^1, \ldots, V_i^T \subseteq P_{i,j}$ for all $i$ such that,

$$\left\| \sum_{\{k,v\} \in P_{i,j}} \exp\left(\frac{\langle k, q_j \rangle}{\sqrt{d}}\right) v - \sum_{l=0}^{T} 2^l \sum_{\{k,v\} \in V_i^l} \exp\left(\frac{\langle k, q_j \rangle}{\sqrt{d}}\right) v \right\|_2$$

$$\leq \varepsilon |P_{i,j}| \cdot e^{-r^2/\sqrt{d}} \cdot 2^i \leq \frac{\varepsilon |P_{i,j}| \cdot 2^i}{\sqrt{j}} \cdot \left\| \exp\left(\frac{K_j \cdot q_j}{\sqrt{d}}\right) \right\|_2,$$

as follows from Theorem 3.4 observing that $\max_{v:\{k,v\} \in P_{i,j}} \|v\|_2 \leq 2^i$. The last inequality holds because $\exp\left(\frac{\langle k, q \rangle}{\sqrt{d}}\right) \geq e^{-r^2/\sqrt{d}}$ and, therefore, $\left\| \exp\left(\frac{K_j \cdot q_j}{\sqrt{d}}\right) \right\|_2 \geq \sqrt{j} \cdot e^{-r^2/\sqrt{d}}$.

Now, observe that $\|V_j\|_F \leq \sqrt{\sum_i |P_{i,j}| \cdot 2^{2i}}$ and $j = \sum_i |P_{i,j}|$. By the Cauchy-Schwartz inequality,

$$\sum_i |P_{i,j}| \cdot 2^i \leq \sqrt{\sum_i |P_{i,j}| \cdot 2^{2i}} \cdot \sqrt{\sum_i |P_{i,j}|}. \tag{5}$$

By triangle inequality, the sum of the outputs of procedures MR-NUMERATOR$_i$ approximates the sum of the terms in the numerator of attention that were not erased in line (8) up to an additive error

$$\frac{\varepsilon}{\sqrt{j}} \cdot \left\| \exp\left(\frac{K_j \cdot q_j}{\sqrt{d}}\right) \right\|_2 \sum_i |P_{i,j}| \cdot 2^i \leq \varepsilon \cdot \left\| \exp\left(\frac{K_j \cdot q_j}{\sqrt{d}}\right) \right\|_2 \|V_j\|_F, \tag{6}$$

where the last inequality follows from Equation (5).

It remains to show how the statement of the theorem follows from the derived bounds. Consider the following abstract derivation. Let $u$ and $u'$ be vectors such that $\|u - u'\|_2 \leq \alpha$, and let $b$ and $b'$ be positive numbers such that

$$\frac{1}{1+\gamma} \cdot \frac{1}{b} \leq \frac{1}{b'} \leq \frac{1}{1-\gamma} \cdot \frac{1}{b}.$$

Then, by application of triangle inequalities,

$$\left\| \frac{u}{b} - \frac{u'}{b'} \right\|_2 \leq \frac{1}{b'} \cdot \|u - u'\|_2 + \|u\|_2 \cdot \left| \frac{1}{b} - \frac{1}{b'} \right|$$

$$\leq \frac{\alpha}{1-\gamma} \cdot \frac{1}{b} + \|u\|_2 \cdot \left(\frac{1}{1-\gamma} - 1\right) \cdot \frac{1}{b} = \tag{7}$$

$$= \frac{\alpha}{1-\gamma} \cdot \frac{1}{b} + \|u\|_2 \cdot \frac{\gamma}{1-\gamma} \cdot \frac{1}{b}.$$

From Equation (4),

$$\frac{1}{1+\varepsilon} \cdot \frac{1}{\sum_{i \in [j]} \exp\left(\frac{\langle k_i, q_j \rangle}{\sqrt{d}}\right)} \leq \frac{1}{\sum_{l=0}^{T} 2^l \sum_{\{k,v\} \in V_i^l} \exp\left(\frac{\langle k, q_j \rangle}{\sqrt{d}}\right)} \leq \frac{1}{1-\varepsilon} \cdot \frac{1}{\sum_{i \in [j]} \exp\left(\frac{\langle k_i, q_j \rangle}{\sqrt{d}}\right)}. \tag{8}$$

For simplicity of notation, let $D \subseteq [j]$ denote the subset of indices of all tokens discarded in line (8). Take $\gamma = \varepsilon, b = \sum_{i \in [j]} \exp\left(\frac{\langle k_i, q_j \rangle}{\sqrt{d}}\right)$, $b' = \sum_{l=0}^{T} 2^l \sum_{\{k,v\} \in V_i^l} \exp\left(\frac{\langle k, q_j \rangle}{\sqrt{d}}\right)$, $u = \sum_{i \in [j] \setminus D} \exp\left(\frac{\langle k_i, q_j \rangle}{\sqrt{d}}\right) v_i$, $u' = \sum_{l=0}^{T} 2^l \sum_{\{k,v\} \in V_i^l} \exp\left(\frac{\langle k, q_j \rangle}{\sqrt{d}}\right) v$ and, finally, $\alpha = \varepsilon \cdot$

$\left\|\exp\left(\frac{K_j \cdot q_j}{\sqrt{d}}\right)\right\|_2 \|V_j\|_F$. The first of the preconditions of our derivation holds by Equation (6). The second one holds by Equation (8). Hence,

$$\left\|\frac{\sum_{l=0}^{T} 2^l \sum_{\{k,v\}\in V^l} \exp\left(\frac{\langle k, q_j\rangle}{\sqrt{d}}\right) v}{\sum_{l=0}^{T} 2^l \sum_{\{k,v\}\in K^l} \exp\left(\frac{\langle k, q_j\rangle}{\sqrt{d}}\right)} - \frac{\sum_{i\in[j]\setminus D} \exp\left(\frac{\langle k_i, q_j\rangle}{\sqrt{d}}\right) v_i}{\sum_{i\in[j]} \exp\left(\frac{\langle k_i, q_j\rangle}{\sqrt{d}}\right)}\right\|_2$$
$$\leq \frac{2\varepsilon}{1-\varepsilon} \cdot \left\|\text{softmax}\left(\frac{K_j \cdot q_j}{\sqrt{d}}\right)\right\|_2 \cdot \|V_j\|_F,$$

since $\|u\|_2 = \left\|\sum_{i\in[j]\setminus D} \exp\left(\frac{\langle k_i, q_j\rangle}{\sqrt{d}}\right) v_i\right\|_2 \leq \left\|\exp\left(\frac{K_j \cdot q_j}{\sqrt{d}}\right)\right\|_2 \|V_j\|_F$.

Finally, combining the above with Equation (3) via triangle inequality, we get

$$\left\|\frac{\sum_{l=0}^{T} 2^l \sum_{\{k,v\}\in V^l} \exp\left(\frac{\langle k, q_j\rangle}{\sqrt{d}}\right) v}{\sum_{l=0}^{T} 2^l \sum_{\{k,v\}\in K^l} \exp\left(\frac{\langle k, q_j\rangle}{\sqrt{d}}\right)} - \text{Attn}(q_j, K_j, V_j)\right\|_2 \leq \frac{2\varepsilon}{1-\varepsilon} \left\|\text{softmax}\left(\frac{K_j \cdot q_j}{\sqrt{d}}\right)\right\|_2 \|V_j\|_F.$$

By rescaling $\varepsilon \to \varepsilon/4$, we get the desired approximation Equation (2). $\square$

## A.2 Theoretical Guarantees of SOFTMAXBALANCE

Algorithm Self-Balancing Walk introduced in [3] receives as input vectors $u_1, \ldots, u_n$ and selects signs for them so that, for any direction, the signed sum of the vectors is balanced along that direction with high probability. The following theorem readily follows from theorem 1.1 in [3]:

**Theorem A.1** (Theorem 1.1 in [3]). *For any $n, d \in \mathbb{N}$, there exists a randomized algorithm which receives as input a set of vectors $U = \{u_1, \ldots, u_n\} \in \mathbb{R}^d$ and a parameter $\delta > 0$. The algorithm outputs a (random) subset $U' \subset U$ such that, for any vector $u \in \mathbb{R}^d$, with probability at least $1 - \delta$,*

$$\left|\sum_{i\in U'} \langle u_i, u\rangle - \sum_{i\notin U'} \langle u_i, u\rangle\right| \leq O\left(\log(n/\delta) \cdot \max_{i\in[n]} \|u_i\|_2 \cdot \|u\|_2\right).$$

---

**Algorithm 3** Self-Balancing Walk $((u_j)_j, r, \delta)$

1: **input:** stream of $\leq n$ vectors $u_j$, radius $r$: $\max_j \|u_j\|_2 \leq r$, probability of failure $\delta$.
2: $c \leftarrow 30 \log(n/\delta)$
3: $U_-, U_+ \leftarrow \emptyset$
4: **for** $i$ from 1 and until the end of the stream **do**
5:     **if** $\left|\sum_{u\in U_+} \langle u, u_i\rangle - \sum_{u\in U_-} \langle u, u_i\rangle\right| > c \cdot r^2$ **then**
6:         Fail
7:     **end if**
8:     $p_i \leftarrow \frac{1}{2} - \frac{\sum_{u\in U_+} \langle u, u_i\rangle - \sum_{u\in U_-} \langle u, u_i\rangle}{2c \cdot r^2}$
9:     $\varepsilon_i \leftarrow +$ with probability $p_i$, and $\varepsilon_i \leftarrow -$ with probability $1 - p_i$
10:     $U_{\varepsilon_i} \leftarrow U_{\varepsilon_i} \cup \{k_i\}$
11: **end for**
12: **if** $|U_+| \leq |U_-|$ **then**
13:     **output:** $U_+$
14: **else**
15:     **output:** $U_-$
16: **end if**

---

*Proof of Theorem 3.3.* Define for any $k \in \mathbb{R}^d$ an embedding function $\varphi(k)$ :

$$\varphi(k) = \left(\frac{(k/d^{0.25})^{\otimes i}}{\sqrt{i!}}\right)_{i\geq 0}.$$

It is easy to see that for any two vectors $k, q \in \mathbb{R}^d$

$$\langle \varphi(k), \varphi(q) \rangle = \exp\left(\frac{\langle k, q \rangle}{\sqrt{d}}\right),$$

and for any $k \in \mathbb{R}^d$

$$\|\varphi(k)\|_2^2 = \exp\left(\frac{\|k\|_2^2}{\sqrt{d}}\right).$$

Consider the set of vectors $\varphi(k_1) \otimes v_1, \ldots, \varphi(k_n) \otimes v_n$. Run the Self-Balancing Walk algorithm on the set of vectors $\varphi(k_1) \otimes v_1, \ldots, \varphi(k_n) \otimes v_n$ with failure parameter set to $\delta/s$ and denote by $C'$ and $C \backslash C'$ the partition of $C$ returned by the algorithm. Observe that, even though vectors $\varphi(k_i) \otimes v_i$ are infinite dimensional, Self-Balancing Walk still can be implemented. The algorithm never has to keep these vectors in the memory because the only operation which requires the knowledge of the embeddings – the inner product – can be performed if we just store vector pairs $\{k_i, v_i\}$:

$$\langle \varphi(k_i) \otimes v_i, \varphi(k_j) \otimes v_j \rangle = \exp\left(\frac{\langle k_i, k_j \rangle}{\sqrt{d}}\right) \cdot \langle v_i, v_j \rangle.$$

Denote by $e_1, \ldots, e_s$ the standard orthonormal basis in $\mathbb{R}^s$. By Theorem A.1, for any $i \in [s]$ with probability $1 - \delta/s$

$$\left| \sum_{\{k,v\} \in C'} \langle \varphi(k) \otimes v, \varphi(q) \otimes e_i \rangle - \sum_{\{k,v\} \notin C'} \langle \varphi(k) \otimes v, \varphi(q) \otimes e_i \rangle \right| \tag{9}$$
$$\leq O\left( \log(ns/\delta) \cdot \max_{\{k,v\} \in C} \|\varphi(k) \otimes v\|_2 \cdot \|\varphi(q) \otimes e_i\|_2 \right),$$

and so with probability at least $1 - \delta$ all of the above inequalities hold simultaneously. To simplify the right hand side, notice that $\|\varphi(k) \otimes v\|_2 = \exp\left(\frac{\|k\|_2^2}{2\sqrt{d}}\right) \cdot \|v\|_2$ and $\|\varphi(q) \otimes e_i\|_2 = \exp\left(\frac{\|q\|_2^2}{2\sqrt{d}}\right)$. Observe that for any $i$ $\langle \varphi(k) \otimes v, \varphi(q) \otimes e_i \rangle = \langle \varphi(k), \varphi(q) \rangle \cdot [v]_i = \exp\left(\frac{\langle k,q \rangle}{\sqrt{d}}\right) \cdot [v]_i$, where by $[v]_i$ we denote the $i$-th coordinate of the vector $v$. Therefore, the left hand side of the expression above is simply the absolute value of the $i$-th coordinate of the vector

$$\sum_{\{k,v\} \in C'} \exp\left(\frac{\langle k, q \rangle}{\sqrt{d}}\right) v - \sum_{\{k,v\} \notin C'} \exp\left(\frac{\langle k, q \rangle}{\sqrt{d}}\right) v.$$

Thus, Equation (9) provides a uniform upper bound on the absolute values of coordinates of the above vector. Since the $l_\infty$ norm of a vector is the maximum of the absolute values of its coordinates,

$$\left\| \sum_{\{k,v\} \in C'} \exp\left(\frac{\langle k, q \rangle}{\sqrt{d}}\right) v - \sum_{\{k,v\} \notin C'} \exp\left(\frac{\langle k, q \rangle}{\sqrt{d}}\right) v \right\|_\infty$$
$$= \max_{i \in [s]} \left| \sum_{\{k,v\} \in C'} \langle \varphi(k) \otimes v, \varphi(q) \otimes e_i \rangle - \sum_{\{k,v\} \notin C'} \langle \varphi(k) \otimes v, \varphi(q) \otimes e_i \rangle \right|$$
$$\leq O\left( \log(ns/\delta) \cdot \max_{\{k,v\} \in C} \left( \exp\left(\frac{\|k\|_2^2}{2\sqrt{d}}\right) \cdot \|v\|_2 \right) \cdot \exp\left(\frac{\|q\|_2^2}{2\sqrt{d}}\right) \right).$$

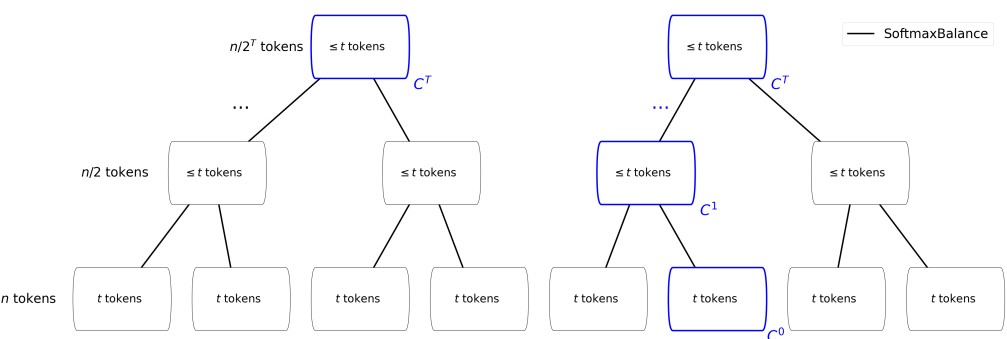

Figure 2: Illustration of the tree structure of MERGEANDREDUCE

Finally, we go from bounding the $l_\infty$ norm a vector to bounding its $l_2$ norm:

$$\left\| \sum_{\{k,v\}\in C'} \exp\left(\frac{\langle k,q\rangle}{\sqrt{d}}\right)v - \sum_{\{k,v\}\notin C'} \exp\left(\frac{\langle k,q\rangle}{\sqrt{d}}\right)v \right\|_2$$

$$\leq \sqrt{s}\left\| \sum_{\{k,v\}\in C'} \exp\left(\frac{\langle k,q\rangle}{\sqrt{d}}\right)v - \sum_{\{k,v\}\notin C'} \exp\left(\frac{\langle k,q\rangle}{\sqrt{d}}\right)v \right\|_\infty$$

$$\leq O\left(\sqrt{s}\cdot \log(ns/\delta)\cdot \max_{\{k,v\}\in C}\left(\exp\left(\frac{\|k\|_2^2}{2\sqrt{d}}\right)\cdot \|v\|_2\right)\cdot \exp\left(\frac{\|q\|_2^2}{2\sqrt{d}}\right)\right).$$

$\square$

## A.3  MERGEANDREDUCE

### A.3.1  Pseudocode for MERGEANDREDUCE

The pseudocode for MERGEANDREDUCE is presented in 4.

---

**Algorithm 4** MERGEANDREDUCE$((k_j, v_j)_j, t, T, \varepsilon)$

---

1: **input:** stream of $\leq n$ tokens $(k_j, v_j)$, batch size $t$, compression rate $2^{-T}$, precision parameter $\varepsilon$.
2: Let SOFTMAXBALANCE be the algorithm as per Theorem 3.3.
3: Initialize $i$-th level subset $C^i, i = 0, \ldots, T$, to empty
4: **repeat**
5:     $C^0 \leftarrow C^0 \cup \{\{k_j, v_j\}\}$
6:     **if** $p$ is not a multiple of $t$ then **then**
7:         **output** $C^0, \ldots, C^T$
8:         **continue**
9:     **end if**
10:     /*Update subsets every $t$ steps*/
11:     $p \leftarrow j/t, i \leftarrow 0$
12:     **while** $p$ is an integer and **until** $i = T$ **do**
13:         $C^{i+1} \leftarrow C^{i+1} \cup$ SOFTMAXBALANCE$(C^i, r_{\text{key}}, r_{\text{value}}, 1/\text{poly}(n))$
14:         $C^i \leftarrow \emptyset$
15:         $i \leftarrow i + 1$
16:         $p \leftarrow p/2$
17:     **end while**
18:     **output** $C^0, \ldots, C^T$
19: **until** token stream ends

---

### A.3.2 Theoretical Guarantees of MERGEANDREDUCE

We now present the full proof of Theorem 3.4.

**Theorem 3.4.** *For any $r, \varepsilon > 0$, any set of tokens $(q_1, k_1, v_1), \ldots, (q_n, k_n, v_n)$ where $q_j, k_j \in \mathbb{R}^d$ satisfy $\|q_j\|_2, \|k_j\|_2 \leq r$, $v_j \in \mathbb{R}^s$ for $s \leq d$ suppose,*

$$\text{batch size } t = \widetilde{O}(\sqrt{s}e^{2r^2/\sqrt{d}}/\varepsilon) \text{ and compression rate } 2^{-T} \text{ with } T = \log(n/t).$$

*Then MERGEANDREDUCE on input parameters $t, r, d, s, \varepsilon$, outputs at every step $j$ of the stream subsets of key-value embedding pairs $C^0, \ldots, C^T \subset C := \{(k_1, v_1), \ldots, (k_n, v_n)\}$ such that, $z_j := \sum_{i=0}^{T} 2^i \sum_{\{k,v\} \in C^i} \exp\left(\frac{\langle k, q_j \rangle}{\sqrt{d}}\right) v$, satisfies with probability at least $1 - 1/poly(n)$,*

$$\left\| \sum_{i=1}^{j} \exp\left(\frac{\langle k_i, q_j \rangle}{\sqrt{d}}\right) v_i - z_j \right\|_2 \leq \varepsilon j \cdot e^{-r^2/\sqrt{d}} \cdot \max_{i \in [n]} \|v_i\|_2.$$

*Total memory of the algorithm is $\widetilde{O}(d\sqrt{s}e^{2r^2/\sqrt{d}}/\varepsilon)$, its $j$-th iteration runtime is $\widetilde{O}(dse^{4r^2/\sqrt{d}}/\varepsilon^2)$.*

*Proof.* Let us first consider the performance of the procedure at time steps which are multiples of $t$. Note that since in the statement of the theorem $T = \log_2(n/t)$, condition **until** in line **while** is redundant. Observe that at any such $j$-th step the procedure is an online implementation of the following simple offline recursive algorithm on dataset $\{\{k_1, v_1\}, \ldots, \{k_j, v_j\}\}$:

1. Set $p = j/t$ and $i = 1$. Split the dataset $\{\{k_1, v_1\}, \ldots, \{k_j, v_j\}\}$ into batches $B_1^0, \ldots, B_p^0$ of size $t$.

2. While $p$ is an integer:

    - Run SOFTMAXBALANCE on the batches $B_1^{i-1}, \ldots, B_p^{i-1}$ independently
    - If $p$ is odd, store the output of SOFTMAXBALANCE on $B_p^{i-1}$ in $C^i$
    - For every $l$, merge the outputs of SOFTMAXBALANCE on $B_{2l-1}^{i-1}$ and $B_{2l}^{i-1}$ into one batch and store them in $B_l^i$,
    - Update $p \leftarrow \lfloor p/2 \rfloor$, $i \leftarrow i + 1$. Stop when $p = 1$.

Therefore, we will analyze space complexity and performance guarantees of the above offline algorithm.

**Probability of success.**

Note that our algorithm performs correctly if each of the calls to SOFTMAXBALANCE produces small error on each of the queries $q$ (as in theorem Theorem 3.3). Throughout the stream, we make $O(n/t)$ calls to SOFTMAXBALANCE, and we apply each to at most $n$ queries, so, it is enough to require that that all SOFTMAXBALANCE have failure probability parameter $\delta = 1/poly(n)$.

**Space complexity.**

Observe that after each iteration of step 2 the number of batches decreases by a factor of two. The maximum batch size is always bounded by $t$. This is because a batch $B_l$ at iteration $i$ of step 2 is a union of SOFTMAXBALANCE$(B_{2l-1})$ and SOFTMAXBALANCE$(B_{2l})$ for $B_{2l-1}$ and $B_{2l}$ at iteration $i - 1$ of step 2, and SOFTMAXBALANCE reduced the size of the dataset which it has been applied to at least by a factor of 2.

The memory of the procedure is the collection of memory cells $C^i$, and $|C^i| \leq t$. Since at time step $j$ at most $\log_2(p) \leq \log_2(n/t)$ memory cells are occupied, the total memory is bounded by $O(dt \log_2(n/t)) = O(dtT)$, which, using the $\widetilde{O}$ notation, is equal to $\widetilde{O}(d\sqrt{d}e^{2r^2/\sqrt{d}}/\varepsilon)$.

**Performance of the algorithm.**

Define $B^i = \cup_l B_l^i$ – the data points which remained in the batches after $i$ iterations of step 2. By triangle inequality,

$$\left\| \sum_{i=1}^{T} 2^i \sum_{\{k,v\} \in C^i} \exp\left(\frac{\langle k, q_j \rangle}{\sqrt{d}}\right) v - \sum_{i=1}^{j} \exp\left(\frac{\langle k_i, q_j \rangle}{\sqrt{d}}\right) v_i \right\|_2$$

$$\leq \sum_{i=0}^{T-1} \left\| 2^{i+1} \sum_{\{k,v\} \in B^{i+1} \cup C^{i+1}} \exp\left(\frac{\langle k, q_j \rangle}{\sqrt{d}}\right) v - 2^i \sum_{\{k,v\} \in B^i} \exp\left(\frac{\langle k, q_j \rangle}{\sqrt{d}}\right) v \right\|_2$$

$$\leq \sum_{i=0}^{T-1} 2^i \left\| \sum_{\{k,v\} \in B^{i+1} \cup C^{i+1}} \exp\left(\frac{\langle k, q_j \rangle}{\sqrt{d}}\right) v - \sum_{\{k,v\} \in B^i \backslash (B^{i+1} \cup C^{i+1})} \exp\left(\frac{\langle k, q_j \rangle}{\sqrt{d}}\right) v \right\|_2.$$

We will refer to the $i$-th summand (starting from 0) on the right hand side as the error produced by the $i + 1$-st iteration of step 2. At the $i + 1$-st iteration of step 2 we apply SOFTMAXBALANCE to $p/2^i$ batches $B_1^i, B_2^i, \ldots$ of size $t$, and we save the outputs of SOFTMAXBALANCE in batches $C^{i+1}, B_1^{i+1}, B_2^{i+1}, \ldots$. Therefore, by Theorem 3.3 and triangle inequality, the error vector produced by the procedure at the $i + 1$-st iteration of step 2 has $l_2$ norm bounded by

$$O\left( 2^i \cdot \sqrt{s} \cdot \log(sn) \cdot \left(\frac{p}{2^i}\right) \cdot e^{r^2/\sqrt{d}} \max_{j \in [n]} \|v_j\|_2 \right) = O\left( \sqrt{s} \cdot \log(sn) \cdot p \cdot e^{r^2/\sqrt{d}} \max_{j \in [n]} \|v_j\|_2 \right),$$

since the error parameter $\delta$ of all instances of SOFTMAXBALANCE is set to $1/\text{poly}(n)$. The $l_2$ norm of the total error of our procedure is bounded by

$$O\left( \sqrt{s} \cdot \log(sn) \cdot T \cdot p \cdot e^{r^2/\sqrt{d}} \max_{j \in [n]} \|v_j\|_2 \right).$$

By definition, $p = j/t$. In order to ensure that the statement of the theorem is correct, the upper bound on the $l_2$ norm of the error vector of the procedure should be less than the desired error $e^{-r^2/\sqrt{d}} \cdot \max_{i \in [n]} \|v_i\|_2$:

$$O\left( \sqrt{s} \cdot \log(sn) \cdot T \cdot \frac{j}{t} \cdot e^{r^2/\sqrt{d}} \max_{j \in [n]} \|v_j\|_2 \right) \leq \varepsilon j \cdot e^{-r^2/\sqrt{d}} \cdot \max_{j \in [n]} \|v_j\|_2.$$

And, since by definition

$$t = O\left( \frac{\log^2(sn) \cdot \sqrt{s} \cdot e^{2r^2/\sqrt{d}}}{\varepsilon} \right),$$

the above inequality holds.

**Runtime during one time step.** At worst, during $j$-th time step the algorithm has to launch SOFTMAXBALANCE $\log_2(p) \leq \log_2(n/t) = T$ times on batches of size $t$, so the runtime is bounded by $O(dt^2 T)$. In the $\widetilde{O}$ notation, the runtime is equal to $\widetilde{O}(d^2 e^{4r^2/\sqrt{d}}/\varepsilon^2)$.

As the final step, we will analyze the performance of the procedure at time steps $j'$ which are not multiples of $t$. Define $j_t = \lfloor j'/t \rfloor \cdot t$. Note that at any such time step the procedure simply saves the triplet $(q_{j'}, k_{j'}, v_{j'})$ and outputs the sum of the approximation $z_{j_t}$ such that

$$\left\| \sum_{i=1}^{j_t} \exp\left(\frac{\langle k_i, q'_j \rangle}{\sqrt{d}}\right) v_i - z_{j_t} \right\|_2 \leq \varepsilon j_t \cdot e^{-r^2/\sqrt{d}} \cdot \max_{i \in [n]} \|v_i\|_2,$$

and $\sum_{i=j_t+1}^{j'} \exp\left(\frac{\langle k_i, q'_j \rangle}{\sqrt{d}}\right) v_i$. From the above inequality,

$$\left\| \sum_{i=1}^{j'} \exp\left( \frac{\langle k_i, q_j' \rangle}{\sqrt{d}} \right) v_i - \left( z_{j_t} + \sum_{i=j_t+1}^{j'} \exp\left( \frac{\langle k_i, q_j' \rangle}{\sqrt{d}} \right) v_i \right) \right\|_2 \leq \varepsilon j_t \cdot e^{-r^2/\sqrt{d}} \cdot \max_{i \in [n]} \|v_i\|_2,$$

as desired. □

# B  Full Experimental Details

Experiments in Section 4.1 and Section 4.3 are performed on a single NVIDIA A100 GPU with 80GB VRAM, and the rest on a single NVIDIA RTX A6000 GPU with 48GB VRAM.

**Implementation Detail.**  To enhance the practical performance of our algorithm, we implement BALANCEKV with parallel operations. Specifically, we consider the cache embeddings of length $n$ and dimension $d$ as a sequence of blocks with length $b$ and reshape them into a tensor of shape $b \times (n/b) \times d$. Then, BALANCEKV is applied in parallel to all blocks of length $b$. For cases where $n$ is not divisible by $b$, we pad the embeddings with zeros. After sign assignment to all embeddings in each block (i.e., line 9 in Algorithm 2), it is reshaped to its original length, and we strictly select $n/2$ embeddings, repeating this process for $T$ iterations.

## B.1  Ablation Studies on Single Layer Attention Approximation

In this section we re-state with full details the single layer attention approximation experiments presented in Section 4.1.

We empirically evaluate the performance of BALANCEKV for approximating a single attention layer, and compare it with independent uniform sampling. We use the pretrained `Llama-3.1-8B-Instruct` [27] and `Ministral-8B-Instruct-2410` [52] and TriviaQA dataset from LongBench [5], and consider the $1^{st}, 2^{nd}$ and $5^{th}$ layers of the models for attention approximation.

For given a prompt with length $n$, we store the corresponding query, key, and value embeddings for all layers. Denote a pair of embeddings in some layer by $(q_1, k_1, v_1), \ldots, (q_n, k_n, v_n)$ and the goal is to approximate the attention $\text{Attn}(q_j, K_j, V_j)$ for the latest 256 queries, i.e. $j \in [n - 256, n]$. Specifically, we keep several first and recent tokens separately and apply BALANCEKV to the intermediate row vectors in $K_j$. This is motivated by StreamingLLM [76] as important contexts are likely contained in the first and latest tokens. We retain the first 256 embeddings and the recent ones from $n - 256$ to $j$ and our compressed cache contains tokens whose indices are in $[256] \cup S \cup \{n - 256, \ldots, j\}$ where $S \subseteq [257, n - 256]$ can be obtained from BALANCEKV. We explore four compression parameters $T \in \{1, 2, 3, 4\}$ which reduces the cache memory by a factor of $2^{-T}$. Let $z_j$ be our approximation using BALANCEKV plus the recent and first few embeddings at the stream $j \in [n - 256, n]$. We compute relative errors $\|z_j - \text{Attn}(q_j, K_j, V_j)\|_F / \|\text{Attn}(q_j, K_j, V_j)\|_F$ for all $j \in [n - 256, n]$, batches, heads and input prompts in the dataset. We repeat this with 10 different random seeds and compute their average and standard deviations. We also compare our method to independent uniform sampling, in which we replace the application of BALANCEKV with sampling a $2^{-T}$ fraction of key and value embeddings with indices in $[257, n - 256]$ uniformly at random. The results are reported in Figure 1.

Next we present the results of the ablation studies described in Section 4.1 which demonstrate how batch size and compression rate affect the relative error in attention approximation for layers 1 and 15 for `Llama-3.1-8B-Instruct`.

## B.2  End-to-End Evaluation on LongBench

We now provide the complete experimental details on the end-to-end evaluation in Section 4.2. We benchmark our algorithm on LongBench dataset [5], a comprehensive collection of datasets designed to evaluate the long-context understanding capabilities of large language models. Specifically, we test a version of uniform length distribution (LongBench-E). The benchmark consists of various long-text application scenarios, including single-document question-answering, multi-document question-answering, summarization, few-shot learning, synthetic tasks and code completion. We use BALANCEKV to compress the key value cache generated in the prefill stage, and maintain all

| Batch Size | 1/2 | 1/4 | 1/8 |
|---|---|---|---|
| 256 | 0.0603 | 0.1190 | 0.1793 |
| 128 | 0.0320 | 0.0624 | 0.0922 |
| 64 | 0.0189 | 0.0349 | 0.0508 |

(a) Layer 1 Runtime (s)

| | 1/2 | 1/4 | 1/8 |
|---|---|---|---|
| 256 | 0.1036 | 0.1764 | 0.2655 |
| 128 | 0.1082 | 0.1833 | 0.2741 |
| 64 | 0.1137 | 0.1921 | 0.2858 |

(b) Layer 1 Relative Error

| | 1/2 | 1/4 | 1/8 |
|---|---|---|---|
| 256 | 0.3920 | 0.4505 | 0.5096 |
| 128 | 0.3654 | 0.3951 | 0.4256 |
| 64 | 0.3592 | 0.3753 | 0.3910 |

(c) Layer 15 Runtime (s)

| | 1/2 | 1/4 | 1/8 |
|---|---|---|---|
| 256 | 0.1107 | 0.1935 | 0.2798 |
| 128 | 0.1121 | 0.1952 | 0.2813 |
| 64 | 0.1141 | 0.1978 | 0.2845 |

(d) Layer 15 Relative Error

Figure 3: Runtime and relative error for across different layers and block sizes. In each figure the rows are corresponding to various batch sizes and columns corresponding to various compression rates

streamed embeddings $(q_j, k_j, v_j)$ during the token decoding/generation stage. This is because the number of generated tokens is much smaller than the input sequence length. We set $b = 256$ and $T = 2$, achieving a consistent compression rate of $0.25$ across all inputs.

We evaluate our method against several token-level key value cache compression schemes, including StreamingLLM [76], SnapKV [46], and PyramidKV [14] as well as uniform sampling described in Section 4.1. We use their implementations from MInference [37], and configure their hyperparameters to match a uniform compression rate with 0.25. We follow the same evaluation metrics from [5]. We test them on `Llama-3.1-8B-Instruct` as well as bigger 14B and 32B parameter models `Qwen-2.5-14B-Instruct` and `Qwen-2.5-32B-Instruct` [77, 71], with results summarized in Table 1.

Our method consistently achieves the highest average performance among compression methods and across all models, demonstrating its effectiveness in preserving model quality for the cache compression. Notably, on the triviaqa dataset, it achieves near-exact scores compared to uncompressed baselines (e.g., 80.68 vs. 81.14 with Qwen2.5-32B), highlighting its ability to retain high-quality information. We observe that uniform sampling performs competitively with our method and this result justifies that a subset obtained from discrepancy theory has practical impacts on various LLM tasks.

### B.3 Needle-In-A-Haystack

In this section we report the plots corresponding to the Needle in a Haystack experiment described in Section 4.3. They are presented in Figure 4.

### B.4 System Efficiency Metrics

In this section we present the prefill and decoding time numbers in Table 2 as described in the system efficiency experimental details in Section 4.4.

| Method | Prefill Time (sec) | Decoding Time (sec) |
|---|---|---|
| Exact | 3.032 | 37.769 |
| SnapKV | 3.755 | 40.426 |
| PyramidKV | 3.748 | 37.241 |
| StreamingLLM | 3.681 | 40.276 |
| BALANCEKV | 3.662 | 38.054 |

Table 2: Minimum wall-clock runtime (in seconds) over 10 trials for prefill and decoding stages.

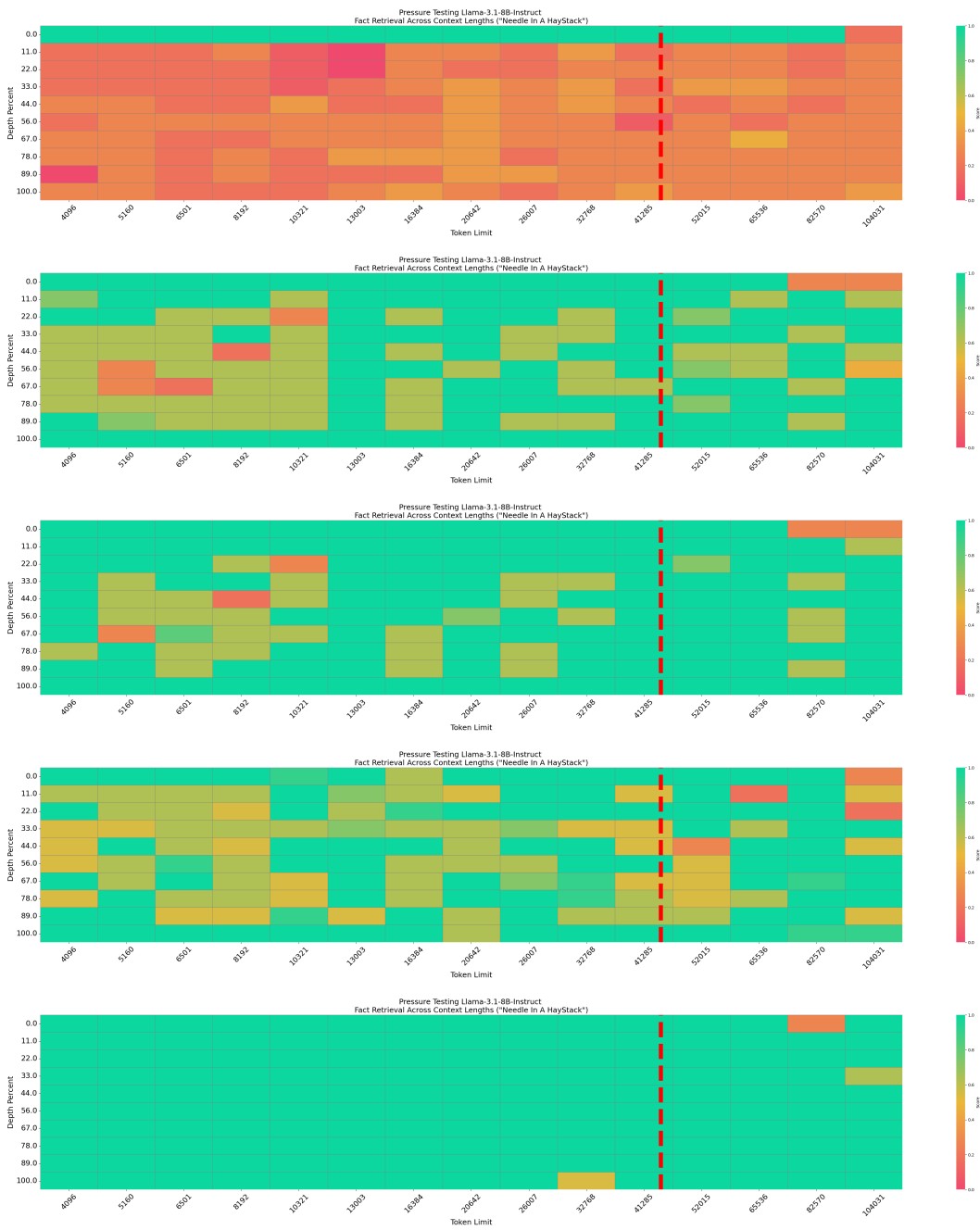

Figure 4: Comparison of performance on Needle in a Haystack task using `Llama-3.1-8B-Instruct`. The methods corresponding to figures from top to bottom are StreamingLLM, SnapKV, PyramidKV, Unif. Sampling and BALANCEKV respectively.

## B.5 Additional Experiments

1. In Table 3, we report the results of the experiment analyzing the $\ell_2$ norms of query (Q), key (K) and value (V) embeddings described in Section 4.5. Due to the space limit, we provide representative results in the below table from (randomly chosen) prompts of various sequence lengths. We note that the reported $\ell_2$ norms of keys shifted by their average, as opposed to the norms of the keys, because our implementation of BALANCEKV shifts the keys by their

average before the compression. It is also easy to see that attention is invariant to the operation of shifting the keys by their average, so it is non-restrictive to assume that the average of the key values is zero.

Table 3: Statistics of norms of qkv embeddings for randomly chosen prompts from TriviaQA.

| Prompt ID | Seq Len | Query | | Key Shifted | | Value | |
|---|---|---|---|---|---|---|---|
| | | Mean | 95% CI | Mean | 95% CI | Mean | 95% CI |
| 10 | 2281 | 14.4512 | 0.0029 | 15.3451 | 0.0065 | 3.3398 | 0.0043 |
| 24 | 3131 | 14.6003 | 0.0025 | 15.6603 | 0.0058 | 3.3539 | 0.0036 |
| 30 | 3388 | 14.7406 | 0.0024 | 15.5152 | 0.0054 | 3.3419 | 0.0035 |
| 22 | 4230 | 14.8339 | 0.0021 | 15.6907 | 0.0049 | 3.3468 | 0.0032 |
| 5 | 5734 | 14.9259 | 0.0019 | 15.7149 | 0.0041 | 3.3482 | 0.0027 |
| 14 | 6616 | 14.9000 | 0.0017 | 15.6757 | 0.0038 | 3.3596 | 0.0025 |
| 4 | 6962 | 14.9743 | 0.0017 | 15.7346 | 0.0039 | 3.3450 | 0.0024 |
| 21 | 8041 | 14.9364 | 0.0016 | 15.7025 | 0.0035 | 3.3491 | 0.0023 |
| 26 | 17337 | 15.1437 | 0.0011 | 15.9531 | 0.0024 | 3.3654 | 0.0016 |
| 27 | 21274 | 15.2065 | 0.0010 | 15.8745 | 0.0022 | 3.3683 | 0.0014 |

2. In Table 4, we report the results of the multimodal task experiment described in Section 4.5.

Table 4: Comparison of BALANCEKV to uniform sampling on MS-COCO, using `InternVL2.5-8B`. The bracket for every method contains the compression rate.

| Method | Bleu_1 | Bleu_2 | Bleu_3 | Bleu_4 | METEOR | RougeL | CIDEr |
|---|---|---|---|---|---|---|---|
| Exact | 0.795 | 0.629 | 0.476 | 0.351 | 0.291 | 0.580 | 1.255 |
| BalanceKV (1/4) | 0.794 | 0.628 | 0.475 | 0.351 | 0.290 | 0.579 | 1.251 |
| Unif (1/4) | 0.794 | 0.629 | 0.476 | 0.350 | 0.290 | 0.578 | 1.247 |
| BalanceKV (1/16) | 0.789 | 0.622 | 0.468 | 0.343 | 0.286 | 0.573 | 1.221 |
| Unif (1/16) | 0.789 | 0.619 | 0.465 | 0.340 | 0.284 | 0.571 | 1.207 |

3. In Table 5, we present the results of the end-to-end experiment on LongBench in the extremely low error regime, described in Section 4.5. We note, that etremely low error regime corresponds to the low compression rate regime, and therefore our experiment is equivalent to exploring the performance of BALANCEKV in the low compression rate regime. For compression rates of 0.8, 0.9 and 0.95, we randomly select a dataset from LongBench and apply both uniform sampling and BALANCEKV to achieve the desired compression rate. More specifically, if we wish to compress a KV cache to $1 - \alpha$ of it's original size, we select its subset of size $2\alpha$, compress it by a factor of 2 with either uniform sampling or BALANCEKV and keep the rest exactly.

Table 5: Comparison of BALANCEKV to uniform sampling in LongBench in the extremely low-error regime.

| Compression Rate | Dataset | BalanceKV | Uniform | Baseline |
|---|---|---|---|---|
| 0.8 | HotpotQA | 50.2 | 48.4 | 51.9 |
| 0.8 | TriviaQA | 91.6 | 86.3 | 91.6 |
| 0.9 | MultiFieldQA | 47.5 | 44.9 | 47.8 |
| 0.9 | Qasper | 42.3 | 39.6 | 43.1 |
| 0.95 | LCC | 49.3 | 45.7 | 49.5 |
| 0.95 | P.Count | 20.7 | 20.1 | 20.7 |

4. In Table 6, we present the results of the end-to-end evaluation of ClusterGen [83] as per the experimental setup in Appendix B.2 and compare its performance to both BALANCEKV and exact attention.

| Method | qasper | multi | hotpotqa | 2wiki | gov | multinews | trec | triviaqa | samsum | p.count | p.ret | lcc | repo-p | average |
|---|---|---|---|---|---|---|---|---|---|---|---|---|---|---|
| **Llama-3.1-8B-Instruct** | | | | | | | | | | | | | | |
| Exact (Baseline) | 42.87 | 48.54 | 52.05 | 38.6 | 31.31 | 22.07 | 71.67 | 91.85 | 42.36 | 20.37 | 98.13 | 49.62 | 42.73 | 50.17 |
| ClusterGen | 33.93 | 42.31 | 50.85 | 37.24 | 21.16 | 19.31 | 67.67 | 90.82 | 39.49 | 20.20 | 96.57 | 47.23 | 39.26 | 46.62 |
| BALANCEKV | 35.75 | 37.04 | 46.37 | 36.24 | 27.09 | 20.84 | 69.0 | 90.88 | 37.88 | 20.39 | 96.65 | 48.45 | 41.4 | **46.77** |

Table 6: Comparison of ClusterGen, BalanceKV and exact attention on LongBench-E using `Llama-3.1-8B-Instruct`. The best results among compression methods for each model are highlighted in bold.

## C  Lower Bound

In this section, we prove the lower bound on the space complexity of an algorithm approximating the $\text{Attn}(\cdot, K, V)$ function. More formally,

**Theorem 3.2.** *Suppose that* $r^2 \leq d$. *Any streaming algorithm which on input* $(\{k_1, v_1\}, \ldots, \{k_n, v_n\}, q)$, $\|q\|_2, \|k_i\|_2 \leq r$, *outputs* $z_q$ *satisfying Equation* (2) *with probability* *0.999 has space complexity* $\Omega\left(\min\{\frac{1}{\varepsilon^2}, d\exp(2r^2/\sqrt{d})\}\right)$.

The proof will be a reduction to the well-known INDEX problem.

### C.1  Reduction to the INDEX Problem

**Definition C.1** (The INDEX problem). Alice gets a bit string $x \sim \text{Unif}\{0, 1\}^n$ and Bob gets $i \sim \text{Unif}[n]$. Then, the goal is to compute $f(x, i) = x_i$ on Bob's end with a single message $m$ from Alice. Denote by $R_\delta^{pub, \rightarrow}$ the public coin one-way communication complexity of computing a function $f(x, y)$ with error probability at most $\delta$: Alice holds $x$, Bob holds $y$, they share a source of random bits and Alice sends a single message to Bob, after which he must output the correct answer with probability at least $1 - \delta$.

**Theorem C.2** (Proven in [75]).

$$R_{2/3}^{pub, \rightarrow}(INDEX) \geq \Omega(n).$$

*Proof of Theorem 3.2.* Let $c$ be the small constant such that $R_{2/3}^{pub, \rightarrow}(INDEX) \geq c \cdot n$.

Assume the contrary to the statement of the Theorem 3.2 – that there exists a streaming algorithm of space complexity $const \cdot \min\{\frac{1}{\varepsilon^2}, d\exp(2r^2/\sqrt{d})\}$ for any sufficiently small constant $const$. We will show that given a string of length $\min\{\frac{1}{\varepsilon^2}, d\exp(2r^2/\sqrt{d})\}$ Alice can solve the INDEX problem as follows. She instantiates such an algorithm with $const < c/C$ for a sufficiently large constant $C$, gives it as input a carefully selected set of keys and values, and sends the state of its memory to Bob. Bob, on his end, can determine whether any randomly drawn bit $i \sim \text{Unif}[n]$ equals 0 or 1 with probability 0.8 by issuing a corresponding (carefully crafted) query to the streaming algorithm and observing its output. Thus, if the streaming algorithm uses small space, we get a contradiction with Theorem C.2, and therefore obtain a proof of Theorem 3.2.

**The reduction.**  Suppose Alice's input to the INDEX problem is a bit string $x \in \{0, 1\}^n$ of length $n = \min\{\frac{1}{\varepsilon^2}, d\exp(2r^2/\sqrt{d})\}$. Using public coins, Alice and Bob jointly generate $n/d$ key vectors $\tilde{k}_1, \ldots, \tilde{k}_{n/d} \sim \text{Unif}\left\{-\frac{r}{\sqrt{d}}, \frac{r}{\sqrt{d}}\right\}^d$, function $\pi: [n] \rightarrow [n/d] \times [d]$ which randomly partitions the $n$ bits into groups of size $d$, and $n$ random signs $\sigma_1, \ldots, \sigma_n \sim \text{Unif}\{-1, 1\}$.

Let $\pi(i)_1 \in [n/d]$ be the first component of $\pi(i)$ and $\pi(i)_2 \in [d]$ – the second component of $\pi(i)$. Let $e_1, \ldots, e_d$ be the standard orthonormal basis in $\mathbb{R}^d$. We build the dataset of $n$ key-value pairs in the following way. We associate with the $i$-th bit the key vector $k_i := \tilde{k}_{\pi(i)_1}$ and the value vector $v_i := \sigma_i \cdot e_{\pi(i)_2}$.

Define

$$U = \{\{k_i, v_i\} : x_i = 1\}$$

the set of key-value pairs corresponding to entries 1 in Alice's bit string $x$. Alice instantiates the streaming algorithm for approximating $\text{Attn}(\cdot, K, V)$ with space complexity $\frac{c}{C} \cdot \min\{\frac{1}{\varepsilon^2}, d \exp(2r^2/\sqrt{d})\}$ with a big enough constant $C$ which we specify later and sends the state of it's memory before reading $q$ to Bob. When Bob receives the message, he uses it to approximate $\text{Attn}(q_i, K, V)$ where $q_i = k_i$. If the value written in the only non-zero coordinate of $v_i$ is larger than $\frac{1}{40} \cdot \frac{\exp(r^2/\sqrt{d})}{\max\{d \exp(r^2/\sqrt{d}), |U|\}}$, Bob reports that the $i$-th bit of Alice's string is equal to 1, and otherwise - 0.

**Analysis of the reduction.**

**Proof sketch.** Before moving to formal proofs we briefly outline the main idea of the analysis. Observe that, by the choice of key vectors, any $\exp(\langle k_j, q_i\rangle/\sqrt{d})$ for $j \neq i$ is in expectation insignificantly small compared to $\exp(\langle k_i, q_i\rangle/\sqrt{d})$ – sometimes we will even refer to these terms as "noise". This statement is formalized in Lemma C.4. Therefore, when Bob computes an approximation to $\text{Attn}(q_i, K, V)$, he will observe a *large* value in the coordinate where $v_i$ is non-zero if $\{k_i, v_i\} \in U$ and a *small* value otherwise.

**Lemma C.3.** $\mathbb{E}_{x,y\sim \text{Unif}\left\{-\frac{r}{\sqrt{d}}, \frac{r}{\sqrt{d}}\right\}^d}[\exp(C\langle x, y\rangle/\sqrt{d})] = \Theta(1)$ *for any constant* $C$.

*Proof.*

$$\mathbb{E}_{x,y\sim \text{Unif}\left\{-\frac{r}{\sqrt{d}}, \frac{r}{\sqrt{d}}\right\}^d}[\exp(C\langle x, y\rangle/\sqrt{d})]$$

$$= \left(\frac{1}{2}\exp(Cr^2/d^{3/2}) + \frac{1}{2}\exp(-Cr^2/d^{3/2}))\right)^d = \cosh\left(\frac{Cr^2}{d^{3/2}}\right)^d,$$

$$1 \leq \exp(C^2r^4/4d^2) \leq \cosh\left(\frac{Cr^2}{d^{3/2}}\right)^d \leq \exp(C^2r^4/d^2) \leq \exp(C^2)$$

where we used the assumption that $r^2/d \leq 1$. $\qquad \square$

**Lemma C.4.** *Fix $i \in [n]$, select $q_i = k_i$. Let $|U_i|$ be the number of key-value pairs in $U \setminus \{k_i, v_i\}$ whose value vector has non-zero entry in the same coordinate as $v_i$.*

*If $\{k_i, v_i\} \in U$ then with probability $> 1 - \frac{1}{1000} \cdot \frac{|U_i|}{|U|}$*

$$\left|\sum_{\{k,v\}\in U} \exp\left(\frac{\langle k, q_i\rangle}{\sqrt{d}}\right)\langle v, v_i\rangle\right| \geq \exp(r^2/\sqrt{d}) - O\left(\sqrt{|U|}\right).$$

*Otherwise, with probability $> 1 - \frac{1}{1000} \cdot \frac{|U_i|}{|U|}$*

$$\left|\sum_{\{k,v\}\in U} \exp\left(\frac{\langle k, q_i\rangle}{\sqrt{d}}\right)\langle v, v_i\rangle\right| \leq O\left(\sqrt{|U|}\right).$$

*Proof.* We prove both statements using Chebyshev's inequality.

In the first case, i.e. when $\{k_i, v_i\} \in U$, the sum contains the term $\exp(\langle k_i, q_i\rangle/\sqrt{d}) = \exp(r^2/\sqrt{d})$, and otherwise it does not. It therefore remains to upper bound the absolute value of the sum

$$X = \sum_{\substack{\{k,v\}\in U, \\ \{k,v\}\neq\{k_i,v_i\}}} \sigma_k \exp\left(\frac{\langle k, q_i\rangle}{\sqrt{d}}\right)\langle v, v_i\rangle = \sum_{\substack{\{k,v\}\in U_i, \\ \{k,v\}\neq\{k_i,v_i\}}} \sigma_k \exp\left(\frac{\langle k, q_i\rangle}{\sqrt{d}}\right)\langle v, v_i\rangle,$$

$\sigma_k \sim \text{Unif}\{-1, 1\}$, which effectively introduces "noise" in Bob's estimate of whether $x_i = 1$. We upper bound this sum now.

$\mathbb{E}[X] = 0$, and $Var(X) = |U_i| \cdot Var_{x,y \sim \text{Unif}\left\{-\frac{r}{\sqrt{d}}, \frac{r}{\sqrt{d}}\right\}^d}(\exp(\langle x, y \rangle / \sqrt{d})) \leq O\left(|U_i|\right)$ because

$$Var_{x,y \sim \text{Unif}\left\{-\frac{r}{\sqrt{d}}, \frac{r}{\sqrt{d}}\right\}^d}(\exp(\langle x, y \rangle / \sqrt{d})) \leq \mathbb{E}_{x,y \sim \text{Unif}\left\{-\frac{r}{\sqrt{d}}, \frac{r}{\sqrt{d}}\right\}^d}[\exp(2\langle x, y \rangle / \sqrt{d})] \leq \exp(4),$$

by Lemma C.3. We therefore get by Chebyshev's inequality

$$\Pr\left[|X| \geq 1000\sqrt{|U|}\right] \leq \frac{1}{1000} \cdot \frac{|U_i|}{|U|}.$$

Therefore, with probability $1 - \frac{1}{1000} \cdot \frac{|U_i|}{|U|}$

$$\left| \sum_{\{k,v\} \in U} \exp\left(\frac{\langle k, q_i \rangle}{\sqrt{d}}\right) \langle v, v_i \rangle \right| \geq \exp(r^2/\sqrt{d}) - 1000\sqrt{|U|}.$$

In the second case, the entire sum equals $X = \sum_{k \in U_i} \sigma_k \exp(\langle k, q_i \rangle / \sqrt{d})$. As shown above, $\Pr\left[|X| \geq 1000\sqrt{|U|}\right] \leq \frac{1}{1000} \cdot \frac{|U_i|}{|U|}$. Hence, with probability $1 - \frac{1}{1000} \cdot \frac{|U_i|}{|U|}$

$$\left| \sum_{\{k,v\} \in U} \exp\left(\frac{\langle k, q_i \rangle}{\sqrt{d}}\right) \langle v, v_i \rangle \right| \leq 1000\sqrt{|U|}.$$

$\square$

**Corollary C.5.** *Suppose bits* $i_1, \ldots, i_d$ *form a group – that is,* $\pi(i_1)_1 = \pi(i_2)_1 = \ldots = \pi(i_d)_1$. *Then all* $v_{i_1}, \ldots, v_{i_d}$ *have different non-zero coordinates, and therefore* $\sum_{j=1}^{d} |U_{i_j}| \leq |U|$.

*Therefore, by the union bound argument, the conclusion of Lemma C.4 holds for all d bits which form one group simultaneously with probability 0.999.*

**Lemma C.6.** *Fix a bit* $i$. *With probability 0.98 the following holds:*

1. *The error of the approximating algorithm in the only non-zero coordinate of* $v_i$ *is bounded by*

$$O\left(\frac{\varepsilon}{\sqrt{d}} \cdot \|softmax(K \cdot q)\|_2 \cdot \|V\|_F\right).$$

2. *If the* $i$-*th bit is 1 then*

$$\left| \sum_{\{k,v\} \in U} \exp\left(\frac{\langle k, q_i \rangle}{\sqrt{d}}\right) \langle v, v_i \rangle \right| \geq \exp(r^2/\sqrt{d}) - O\left(\frac{\sqrt{|U|}}{\sqrt{d}}\right),$$

   *and*

$$\left| \sum_{\{k,v\} \in U} \exp\left(\frac{\langle k, q_i \rangle}{\sqrt{d}}\right) \langle v, v_i \rangle \right| \leq O\left(\frac{\sqrt{|U|}}{\sqrt{d}}\right).$$

   *otherwise.*

*Proof.* We may think that the process of generating the dataset and the approximating streaming algorithm has the following order: first Alice and Bob jointly generate the partition $\pi$, the key vectors $\tilde{k}_1, \ldots, \tilde{k}_{n/d}$ and the value vectors $v_1, \ldots, v_n$ all using public randomness. To generate Bob's input position $i \in [n]$ we generate pair $a \sim \text{Unif}[n/d]$, $b \sim \text{Unif}[d]$ and declare $i = \pi^{-1}(a, b)$. We may assume that $a$ is chosen before the datasets $K$ and $V$ are generated, and $b$ – after.

Before $b$ is drawn, the key vector $k_i$ of $i = \pi^{-1}(a, b)$ is already defined, as well as the datasets $K, V$ and $U$. Alice can therefore already apply the streaming algorithm to $U$, and Bob can already apply it to $q_i = k_i$. Therefore, the error vector which the streaming algorithm yields when applied to $q_i = k_i$ is also defined before $b$ is known.

Clearly, there are no more than $0.0001 \cdot d$ coordinates in which the error of approximation exceeds $10000 \cdot \frac{\varepsilon}{\sqrt{d}} \cdot \|\text{softmax}(K \cdot q)\|_2 \cdot \|V\|_F$. Since every value vector has only one non-zero entry, there are no more than $0.0001 \cdot d$ coordinates where at least $10000 \cdot \frac{U}{d}$ of value vectors from $U$ have non-zero value. We call all coordinates which are in neither of these two groups *safe*. From the above, at least 99% of the coordinates are safe. Recall that $b \sim \text{Unif}[d]$, and choosing $b$ is equivalent to choosing the coordinate in which $v_i$ is non-zero. Therefore, with probability 0.99 over the choice of $b$ the only non-zero coordinate of $v_i$ is safe.

At the same time, similarly to Lemma C.4, by Chebyshev inequality, if $U_i \subset U$ is the set of all key-value pairs in $U$ whose value vector has the same non-zero coordinate as $v_i$ then with probability 0.999 if the $i$-th bit is 1 then

$$\left| \sum_{\{k,v\} \in U} \exp\left( \frac{\langle k, q_i \rangle}{\sqrt{d}} \right) \langle v, v_i \rangle \right| \geq \exp(r^2/\sqrt{d}) - O\left( \sqrt{|U_i|} \right),$$

and

$$\left| \sum_{\{k,v\} \in U} \exp\left( \frac{\langle k, q_i \rangle}{\sqrt{d}} \right) \langle v, v_i \rangle \right| \leq O\left( \sqrt{|U_i|} \right).$$

otherwise.

By union bounding over these two events, we get that the statement of the lemma is correct with high constant probability.

$\square$

*Conclusion of the proof.* Let $U^i \subset U$ be the set of all pairs from $U$ with the same key as $\{k_i, v_i\}$. Since Alice's string is drawn from $\text{Unif}\{-1, 1\}^n$, with probability 0.9 $|U^i| \geq 0.4 \cdot d$. This is because every bit in the same group as $k_i$ belongs to $U$ with probability 1/2.

Observe that by Chebyshev inequality, with high probability 0.999, the denominator of $\text{softmax}(K \cdot q_i)$ lies in range

$$\left[ |U^i| \cdot \exp(r^2/\sqrt{d}) + \frac{1}{5} \cdot |U|, |U^i| \cdot \exp(r^2/\sqrt{d}) + 20 \cdot |U| \right],$$

This is because every summand in the denominator, except for $\exp(\langle k_i, q_i \rangle / \sqrt{d})$, is distributed as $\exp(\langle x, y \rangle / \sqrt{d})$, $x, y \sim \text{Unif}\left\{ -\frac{r}{\sqrt{d}}, \frac{r}{\sqrt{d}} \right\}^d$, and the expectation and the variance of this distribution, as shown in Lemma C.3, is $\Theta(1)$. This range is contained in $\left[ \frac{1}{5} \cdot (\max\{de^{r^2/\sqrt{d}}, |U|\}), 20 \cdot (\max\{de^{r^2/\sqrt{d}}, |U|\}) \right]$. We will denote the denominator as $D$.

Similarly, by Chebyshev inequality, with probability 0.999 the numerator of $\text{softmax}(K \cdot q_i)$ lies in

$$\left[ \sqrt{|U^i| \cdot \exp(2r^2/\sqrt{d}) + \frac{1}{5} \cdot |U|}, \sqrt{|U^i| \cdot \exp(2r^2/\sqrt{d}) + 200 \cdot |U|} \right]$$

which, since $|U| \leq d \exp(2r^2/\sqrt{d})$, is bounded by $\sqrt{200} \cdot \sqrt{d} e^{r^2/\sqrt{d}}$.

Suppose that the $i$-th bit is 1. Then,

- When $\frac{1}{\varepsilon} \geq \sqrt{d}e^{r^2/\sqrt{d}}$, by selecting $|U| = \frac{c}{C} \cdot de^{2r^2/\sqrt{d}}$ for some enough constant $C$ the value written in the only non-zero coordinate of $v_i$ is at least $\frac{e^{r^2/\sqrt{d}}}{D} - \frac{1}{100}\frac{e^{r^2/\sqrt{d}}}{D}$ and at most $\frac{1}{100}\frac{e^{r^2/\sqrt{d}}}{D}$ otherwise, as follows from Lemma C.4;

- When $\frac{1}{\varepsilon} < \sqrt{d}e^{r^2/\sqrt{d}}$, by selecting $|U| = \frac{c}{C} \cdot \frac{1}{\varepsilon^2}$ for some big enough constant $C$ the value written in the only non-zero coordinate of $v_i$ is at least $\frac{e^{r^2/\sqrt{d}}}{D} - \frac{1}{100} \cdot \frac{1}{\varepsilon \cdot D \cdot \sqrt{d}}$, and at most $\frac{1}{100} \cdot \frac{1}{\varepsilon \cdot D \cdot \sqrt{d}}$ otherwise, as follows from Lemma C.4.

The error which the approximator can have in the non-zero coordinate of $v_i$ is bounded by

$$10000\frac{\varepsilon}{\sqrt{d}} \cdot \|\text{softmax}(K \cdot q)\|_2 \cdot \|V\|_F \leq 10000\frac{\varepsilon}{\sqrt{d}} \cdot \frac{\sqrt{20} \cdot \sqrt{d}e^{r^2/\sqrt{d}}}{D} \cdot \sqrt{|U|},$$

as shown in Lemma C.6. Below, we show that this error is smaller than the gap between $\frac{1}{40} \cdot \frac{e^{r^2/\sqrt{d}}}{\max\{de^{r^2/\sqrt{d}}, |U|\}}$ and the value written in the coordinate, which means that, even though the approximator introduces some error, Bob is still capable to tell whether the $i$-th bit is 1 or 0.

- When $\frac{1}{\varepsilon} \geq \sqrt{d}e^{r^2/\sqrt{d}}$, the gap between the value written in the coordinate and $\frac{1}{40}\frac{e^{r^2/\sqrt{d}}}{\max\{de^{r^2/\sqrt{d}}, |U|\}}$ is at least $\frac{1}{1000} \cdot \frac{e^{r^2/\sqrt{d}}}{\max\{de^{r^2/\sqrt{d}}, |U|\}}$, and the error

$$\frac{10000\varepsilon}{\sqrt{d}} \cdot \frac{\sqrt{20} \cdot \sqrt{d}e^{r^2/\sqrt{d}}}{D} \cdot \sqrt{|U|} \leq \frac{\varepsilon}{2000} \cdot \frac{\sqrt{d}\exp(2r^2/\sqrt{d})}{\max\{de^{r^2/\sqrt{d}}, |U|\}} \leq \frac{1}{2000} \cdot \frac{e^{r^2/\sqrt{d}}}{\max\{de^{r^2/\sqrt{d}}, |U|\}}$$

by an appropriate choice of $C$.

- When $\frac{1}{\varepsilon} < \sqrt{d}e^{r^2/\sqrt{d}}$, the gap between the value written in the coordinate and $\frac{1}{40}\frac{e^{r^2/\sqrt{d}}}{\max\{de^{r^2/\sqrt{d}}, |U|\}}$ is at least $\frac{1}{1000} \cdot \frac{e^{r^2/\sqrt{d}}}{\max\{de^{r^2/\sqrt{d}}, |U|\}}$ and the error

$$\frac{10000\varepsilon}{\sqrt{d}} \cdot \frac{\sqrt{20} \cdot \sqrt{d}e^{r^2/\sqrt{d}}}{D} \cdot \sqrt{|U|} \leq \frac{\varepsilon}{2000} \cdot \frac{\exp(r^2/\sqrt{d})}{\max\{de^{r^2/\sqrt{d}}, |U|\} \cdot \varepsilon} \leq \frac{1}{2000} \cdot \frac{\exp(r^2/\sqrt{d})}{\max\{de^{r^2/\sqrt{d}}, |U|\}}$$

by an appropriate choice of $C$.

$\square$

