# OpenReview forum: "Streaming Attention Approximation via Discrepancy Theory"
_NeurIPS.cc/2025/Conference — NeurIPS 2025 spotlight_

### Official Review · Reviewer_CpPu · 2025-06-21

**Clarity:** 4
**Significance:** 3
**Originality:** 3
**Rating:** 5
**Confidence:** 4

**Summary:**

This paper proposes a theoretically principled KV cache compression mechanism based on recursively applying the self-balancing walk algorithm for vector discrepancy minimization. The method achieves some improvements in terms of efficiency and model quality when compared to previous techniques.

**Questions:**

1. I have not heard of the term "streaming complexity" before? I suppose it refers to space complexity?
2. Line 238 typo: "It's as"
3. In Theorem 3.4, the guarantee is "for any vector" $q \in \mathbb{R}^d$. However, we are only concerned with the embeddings $q_1,...,q_n$. Is there any room for improvement in the context of the self-balancing walk algorithm if you have this relaxation?
4. What if the number of generated tokens is not known in advance? How would you set $T$?
5. When calculating the denominator to the softmax, you are providing a vector of all $1$-s to the Balancing algorithm. That value vector has $\ell_2$ norm equal to $\sqrt{s}$. Is that accounted for in the analysis?
6. In the proof of Theorem 3.4, the argument is that the self-balancing algorithm can be performed on the infinite-dimensional vectors generated by applying the exponential kernel $\phi$. That I believe is correct, but I think some care is needed about arguing that the radius of these vectors is, in some way, bounded in terms of $r$. This seems to be a prerequisite to correctly applying Theorem A.1, and I didn't see an argument for it anywhere. Is it obvious perhaps?
7. What is the "prefill" stage? It is not clear what this stage is architecturally in the experiments you are mentioning.

**Ethical Concerns:**

["NO or VERY MINOR ethics concerns only"]

**Final Justification:**

The paper contains a very nice algorithm for KV cache compression based on discrepancy minimization. The theoretical contributions are very nice and solid, and the experiments performed are insightful and promising. Few small revisions and clarifications in the main text are required to improve clarity, as per the discussions.

**Limitations:**

Yes, though I believe that a big more extensive discussion on limitations and open questions would be a good addition to the paper.

**Quality:**

4

**Strengths And Weaknesses:**

### Strengths
1. The paper is well-written and motivated. I found the presentation flowing very well, and the text is written clearly and methodically.
2. The paper is timely positioned and contributes to an important problem - the memory complexity of inference in attention-based autoregressive transformers.
3. The approach is novel and backed by very nice theory. I found the application of discrepancy minimization algorithms very neat in this context. The paper does a good job of linking this important theoretical problem to this new domain and adapting it to its needs. Theoretically the contribution is adapting the existing method of discrepancy minimization to attention and not developing the theory anew, which the author carefully clarify, placing the paper in the correct context and not over-stating its contribution.
4. The experiments performed are thorough and (for the most part) reflective of the statements the theory is trying to support. The authors experiment both with synthetic transformer layers and with real models, noting their method's performance over prior work.
5. The lower bound contained in the appendix is a nice complement to this set of results.

### Weaknesses
There aren't a lot of weaknesses, but I have a few items of note:
1. I find the related work section a bit limited. I think it would be a good idea to add (possibly to the appendix) a few more related works on efficient attention computation, KV cache compression methods and theory for sublinear algorithms on attention, of which there is a lot. For example, there have been a few works on the sub-quadratic (offline) attention and hardness (eg. [1], [2] etc...), that, while not addressing the online streaming setting, would make for a comprehensive literature review.
2. In the same vein, there is also a recent work [3] that focuses on lower bounds for KV cache compression. Their results appear complimentary to yours - specifically their lower bounds study the dependence on the stream length, where as yours study the dependency on the norms of the input vectors. It would be nice to discuss this briefly also in the related works section.
3. For the experiments where you tested your method in "real" LLMs, what layers did you apply compression to? Did you notice a significant deterioration in model quality when the number of layers increases? My intuition says that as the number of layers increase your error will accumulate, so even if one layer is approximated well the net effect will be model degradation. Have you studied how this phenomenon appears?
4. I'm not sure about the benefits of comparing with uniform sampling. Uniform sampling seems like it would work only in specific settings. Did you try comparing with Subgen [4]? It seems that this is another algorithm with theoretical guarantees for streaming attention.
5. There seem to be a handful of assumptions involving the norms of the embeddings involved. Have you checked if those are generally satisfied in practice? Some comments on this would be important because the algorithm's memory depends exponentially on the square of the spectral radius $r$. I do believe that that quantity should, in general, be small, but it takes a bit away from the purity of the result. Of course, the lower bound shows the complexity is tight in that aspect, so this is not a weakness per se. Also, the error is also expressed in terms of the Frobenius norm of the value embedding matrix, which could, reasonably be large.
6. I think a more extensive discussion of limitations and open questions left behind by the findings of this work would be a good addition to the paper overall. Currently such a discussion is very limited.

[1] Alman, Josh, and Hantao Yu. "Fundamental Limitations on Subquadratic Alternatives to Transformers." arXiv preprint arXiv:2410.04271 (2024).

[2] Alman, Josh, and Zhao Song. "Fast attention requires bounded entries." Advances in Neural Information Processing Systems 36 (2023): 63117-63135.

[3] Haris, Themistoklis, and Krzysztof Onak. "Compression barriers for autoregressive transformers." arXiv preprint arXiv:2502.15955 (2025).

[4] Zandieh, Amir, et al. "Subgen: Token generation in sublinear time and memory." arXiv preprint arXiv:2402.06082 (2024).

---

> ### Author Rebuttal · Authors · 2025-07-31
>
> Thank you for carefully studying our submission and the additional, very useful suggestions and comments!
>
> -->I find the related work section a bit limited. I think it would be a good idea to add (possibly to the appendix) a few more related works on efficient attention computation, KV cache compression methods and theory for sublinear algorithms on attention, of which there is a lot. In the same vein, there is also a recent work [3] that focuses on lower bounds for KV cache compression. Their results appear complimentary to yours - specifically their lower bounds study the dependence on the stream length, where as yours study the dependency on the norms of the input vectors. It would be nice to discuss this briefly also in the related works section…I think a more extensive discussion of limitations and open questions left behind by the findings of this work would be a good addition to the paper overall. Currently such a discussion is very limited.
>
> Response: Thanks a lot for pointing out this as well as related work on efficient attention and kv cache compression, and lower bounds. We will incorporate these and add relevant discussion of the limitations of this work and future directions in the final version of the paper.
>
> -->For the experiments where you tested your method in "real" LLMs, what layers did you apply compression to? Did you notice a significant deterioration in model quality when the number of layers increases? My intuition says that as the number of layers increase your error will accumulate, so even if one layer is approximated well the net effect will be model degradation. Have you studied how this phenomenon appears?
>
> Response: In our end to end experiments we apply the compression to all layers. Regarding the latter part of your question, we perform the following experiment - we evaluate the end to end performance of Llama 3.1 8B Instruct after applying the compression on only the first i layers on Qasper (one of the datasets in LongBench), for the values i in [0,8,16,24,32] (32 is the last layer), and observe the scores of [42.8,42.4,39.8,.38.2,35.8] respectively. This confirms your intuition that as the number of layers on which the compression is applied increases, the performance decreases.
>
> -->I'm not sure about the benefits of comparing with uniform sampling. Uniform sampling seems like it would work only in specific settings. Did you try comparing with Subgen [4]? It seems that this is another algorithm with theoretical guarantees for streaming attention.
>
> Response: The fundamental goal of the paper is to demonstrate the power of the geometric sampling in streaming attention approximation. Therefore, geometric sampling is compared to uniform sampling as baseline.
>
> We further benchmark Subgen [4] and compare it to our BalanceKV using Llama-3.1 model and a subset of LongBench datasets (due to limited timeline). The experiments were performed with a single NVIDIA RTX A6000 GPU with 48GB VRAM. The below table summarizes the performance of each method on datasets we test.
>
> |               Method | qasper  | multifieldqa_en | gov_port | multnews | trec   | triviaqa |
> |----------------------|---------|------------------|----------|-----------|--------|----------|
> | Exact                | 42.87  | 48.54           | 31.31   | 22.07    | 71.67 | 91.85   |
> | Subgen             | 33.93  | 42.31           | 21.16   | 19.31    | 67.67 | 90.82   |
> | BalanceKV       | 35.75  | 37.04           | 27.09   | 20.84    | 69.00 | 90.88   |
>
> As a result, our BalanceKV outperforms Subgen on all datasets but “multifieldqa_en”. We will add complete experiments in our final draft.
>
> -->There seem to be a handful of assumptions involving the norms of the embeddings involved. Have you checked if those are generally satisfied in practice? Some comments on this would be important because the algorithm's memory depends exponentially on the square of the spectral radius . I do believe that that quantity should, in general, be small, but it takes a bit away from the purity of the result. Of course, the lower bound shows the complexity is tight in that aspect, so this is not a weakness per se. Also, the error is also expressed in terms of the Frobenius norm of the value embedding matrix, which could, reasonably be large.
>
>
> Response: Thanks for the question. Please refer to point 2 of the response to Reviewer BB75 for the discussion of the upper bounds on the norms of the embedding vectors.
> Regarding the choice of the Frobenius norm - this is a great suggestion and it is a very interesting open problem to obtain the same space bounds as ours, but with error guarantee scaling with the spectral norm of V. Note that the inequality $||V||_F \leq \sqrt{d}||V||_2 $implies that by rescaling the epsilon we can use BalanceKV to get an algorithm with the standard error of attention approximation, whose space and time complexity is scaled by $\sqrt{d}$ and $d$ respectively. Furthermore, we observe in practice that the singular values of the matrix V decay fast, thus the theoretical upper bound of $\sqrt{d}||V||_2$ on $||V||_2$ is loose.
>
> Questions:
>
> 1. I have not heard of the term "streaming complexity" before? I suppose it refers to space complexity?
>
> Response: You are absolutely correct, we refer to the space complexity of a streaming algorithm solving the problem of attention approximation when we write ``streaming complexity of attention approximation’’.
>
> 2. In Theorem 3.4, the guarantee is "for any vector". However, we are only concerned with the embeddings $q_1, \ldots, q_n$. Is there any room for improvement in the context of the self-balancing walk algorithm if you have this relaxation?
>
> Response: Indeed, the query embeddings definitely have special properties; this is evidenced, for example, by the existence of attention sinks (https://arxiv.org/pdf/2309.17453). In more detail, it has been observed that the initial input tokens have large attention scores, which, in particular, suggests that the query embeddings must be non-trivially correlated with the key embeddings of the initial input tokens. It is an exciting direction for future work to discover more geometric properties of the query embeddings and leverage them to improve BalanceKV.
>
> 3. What if the number of generated tokens is not known in advance? How would you set T?
>
> Response: In practice, the maximum number of generated tokens is a property of the model and, therefore, it is known in advance.
>
> 4. When calculating the denominator to the softmax, you are providing a vector of all 1-s to the Balancing algorithm. That value vector has $\ell_2$ norm equal to $\sqrt{s}$. Is that accounted for in the analysis?
>
> Response: When we compute the approximation to the denominator we set all of the value vectors trivially equal to scalar 1. In other words, they are indeed vectors of all 1-s but the dimension, s, is 1.
>  To briefly justify this choice of value vectors, note that Theorem 3.4 proves theoretical guarantees of SoftmaxBalance which may be applied to approximate $\sum \exp(\langle k, q\rangle/\sqrt{d})v$ where the dimension of the value vectors is arbitrary. Since both the numerator and the denominator of the attention function may be expressed in this way, we approximate both with the same algorithm, just applied to different sets of value vectors. Setting all value vectors to be scalars 1, the expression  $\sum \exp(\langle k, q\rangle/\sqrt{d})v$ becomes precisely the denominator of the attention function.
>
> 5. In the proof of Theorem 3.4, the argument is that the self-balancing algorithm can be performed on the infinite-dimensional vectors generated by applying the exponential kernel $\phi$. That I believe is correct, but I think some care is needed about arguing that the radius of these vectors is, in some way, bounded in terms of r. This seems to be a prerequisite to correctly applying Theorem A.1, and I didn't see an argument for it anywhere. Is it obvious perhaps?
>
> Response: Indeed, we apply Theorem A.1 to vectors $\varphi(k)$ and $\varphi(q)$, and the guarantees of Theorem A.1 depend on the $\ell_2$ norms of these vectors. In the expression in between lines 918 and 919 we note that $||\varphi(k)||^2_2 = \exp(||k||^2_2/\sqrt{d})$.
> Ultimately, we use our assumption $||k||_2, ||q||_2 \leq r$, which gives the desired upper bound on the norms of the infinite dimensional vectors. Note, however, that Theorem A.1 is only used in the proof of Theorem 3.4, which does not assume anything about boundedness of norms of key and query vectors and whose guarantees are stated in terms of max $||k||^2_2$. This is why we did not mention r in that proof. We will add a discussion regarding this in the final version, to improve the understanding of the big picture.
>
> 6. What is the "prefill" stage? It is not clear what this stage is architecturally in the experiments you are mentioning.
>
> Response: ''Prefill'' refers to the initial forward pass over the entire input prompt to populate the model’s KV (key-value) cache. During the next stage, referred to as ''generation'' or ''decoding'', the new tokens are generated one by one, and in this process it is crucial to compute the attention function between query embeddings of each newly generated token and key value embeddings of each past token, which implies the need to keep the entire KV cache during ''decoding''.
> In our experiments, we do not optimize the ''prefill'' stage and compute the exact KV cache with standard methods. However, during the ''prefill'' stage we run the BalanceKV algorithm to compress the exact KV cache at every head and every layer to decrease the space complexity during the ``decoding’’ stage.

---

> > ### Comment · Reviewer_CpPu · 2025-08-03
> > **Response to Rebuttal**
> >
> > Thank you for your clarifications. I will maintain my score.

---

### Official Review · Reviewer_VYzU · 2025-06-23

**Clarity:** 3
**Significance:** 3
**Originality:** 3
**Rating:** 5
**Confidence:** 3

**Summary:**

In this paper, the authors present a new approach to compress the KV cache in LLMs. This addresses the memory bottleneck in long context token generation. This novel technique is based on a sampling process from Banaszczyk's vector balancing theory. The authors use discrepancy theory to intelligently select which key-value pairs to retain in the cache rather than using simple uniform sampling or heuristic-based approaches.

**Questions:**

Please see weaknesses

**Ethical Concerns:**

["NO or VERY MINOR ethics concerns only"]

**Final Justification:**

I thank the authors for providing these clarifications, which have addressed all of my concerns. Since my initial rating was already high, I choose to maintain my original score.

**Limitations:**

Yes

**Quality:**

3

**Strengths And Weaknesses:**

The strengths of this paper can be summarized as follows:

First, this paper has a very rigorous theoretical analysis for both the upper and lower bounds. As far as I can see, the theoretical analyses are all correct. Using Banaszczyk's vector balancing theory to attention approximation is very creative, and this is well-motivated.

Second, the experimental results seem convincing. It covers multiple tasks, such as single-layer approximation, end-to-end performance, needle-in-a-haystack tasks, and system efficiency metrics. This makes the result very influential.

The weaknesses of this paper can be summarized as follows:

First, the theoretical result of this paper requires the assumption that the norms of the query and keys are bounded. This might not hold in all the practical situations. Can the authors briefly discuss when this will hold?

Second, it seems that this method requires tuning multiple hyperparameters, like batch size and compression rate, as shown in Theorem 3.1. It would be helpful if the authors could provide clear guidance on hyperparameter selection for different scenarios.

---

> ### Author Rebuttal · Authors · 2025-07-31
>
> Thank you for carefully studying our submission and the additional, very useful suggestions and comments!
>
> 1. First, the theoretical result of this paper requires the assumption that the norms of the query and keys are bounded. This might not hold in all the practical situations. Can the authors briefly discuss when this will hold?
>
> Response: Thanks for the question, please refer to point 2 of the response to Reviewer BB75 where this is discussed.
>
>
> 2. Second, it seems that this method requires tuning multiple hyperparameters, like batch size and compression rate, as shown in Theorem 3.1. It would be helpful if the authors could provide clear guidance on hyperparameter selection for different scenarios.
>
> Response: The compression rate is, in practice, dictated by the memory budget. As we discuss in the response to question 1 of Reviewer pBkK, there is only one hyperparameter in our experiments which requires tuning - the parameter gamma which is a function of the upper bound on the norms of key and query vectors. We observed that quite a wide range of choices for this parameter lead to similar performance in practice, and we fix it to be 4 across all experiments.

---

> > ### Comment · Reviewer_VYzU · 2025-08-03
> >
> > Thank you very much for the clarifications. I have no further questions and will maintain my original score.

---

### Official Review · Reviewer_w4SJ · 2025-06-30

**Clarity:** 3
**Significance:** 3
**Originality:** 3
**Rating:** 4
**Confidence:** 3

**Summary:**

The use of KV cache in autoregressive transformers requires storing all the keys and values, which inevitably takes $O(nd)$ memory, where $n$ is the context length and $d$ is the embedding dimension. This paper uses techniques in streaming algorithms and discrepancy theory to design an algorithm that only uses approximately $O(d^{1.5}\exp(2r^2/\sqrt{d}))$ memory (suppressing $\log n$ factors), where $r$ is an upper bound on the $\ell_2$ norm of queries and keys. An almost matching lower bound is proved additionally. The theory is further verified with experiments where they compare their techniques with previous ones.

The main technical ideas are:
1. For each query $q_i$, we want to partition $[i]$ into two sets such that the weighted sum of the values are close, and recurse on the smaller set. Based on the self-balancing walk algorithm (which was designed for the inner product function), this paper develops an algorithm for this problem with respect to $\exp(\langle k,\cdot\rangle/\sqrt{d})$ function.
2. Furthermore, this paper deals with the streaming setup by partitioning the stream of tokens into batches and perform the algorithm above in each batch (merge and reduce).

**Questions:**

1. I don’t quite understand line 186 - was the previous SOTA algorithm using around $d/\varepsilon^2$ memory? Or is the trivial $nd$ memory the previous SOTA? I am not sure whether improving from $d/\varepsilon^2$ to $d^{1.5}/\varepsilon$ is a big improvement.
2. When the recursive algorithm (softmaxbalance) is performed, is the error also accumulating exponentially? Eventually the error will depend on the error between the two sets in each partition.
3. I did not check the appendix, but is the algorithm overall easy to implement? It seems like there will be a lot of extra bookkeeping that might cause more memory usage for larger models.

**Ethical Concerns:**

["NO or VERY MINOR ethics concerns only"]

**Final Justification:**

This paper provides an interesting way to save memory usage in transformers. The theory results are interesting and further verified by some experiments, although their theoretical techniques have been extensively used in theoretical computer science (for different purposes). I recommend a borderline accept for these reasons.

**Limitations:**

yes

**Quality:**

3

**Strengths And Weaknesses:**

Strengths:
1. Memory is a major bottleneck for transformers, so studying more efficient ways to store keys, queries and values is an important research area for transformers.
2. The problem setup, high-level ideas are well explained and written, especially section 2.
3. The empirical results solidify the theory results and illustrate that BalancedKV outperforms previous techniques on several benchmark datasets.

Weaknesses:
1. The techniques presented, self-balancing walk algorithm and merge and reduce, have both been popular techniques that are widely used. There seems to be less novelty because of this.
2. What is a good estimation of $r$ in practice? A large $r$ will weaken the results.
3. This paper gives error upper bound for a single attention layer, but the error could compound when we have more heads and layers?

---

> ### Author Rebuttal · Authors · 2025-07-31
>
> Thank you for carefully studying our submission and the additional, very useful suggestions and comments!
>
> -->What is a good estimation of r in practice? A large r will weaken the results.
>
> Response: Thank you for pointing this out. Please refer to the 2nd point of the response to Reviewer BB75 for the discussion of the upper bounds on the norms of keys and queries in the real data.
>
> -->This paper gives error upper bound for a single attention layer, but the error could compound when we have more heads and layers?
>
> Response: Thank you for this observation! Indeed, the error of BalanceKV could compound across multiple layers. We show this does not lead to significant performance degradation in practice - in our end-to-end experiments we show applying BalanceKV across all layers and heads leads to better performance as compared to other state of the art methods.
>
> Questions:
>
> 1. I don’t quite understand line 186 - was the previous SOTA algorithm using $d/\epsilon^2$ memory? Or is the trivial  memory nd the previous SOTA? I am not sure whether improving from $d/\epsilon^2$ to $d^{1.5}/\epsilon$ is a big improvement.
>
> Response: Thank you for your question! Indeed, uniform sampling has the theoretical guarantee of $d/\epsilon^2$  and is the previous SOTA. From the theoretical perspective, BalanceKV outperforms uniform sampling in the low error regime: more precisely, when $\epsilon < 1/d$. However, as our experiments show, in practice BalanceKV outperforms uniform sampling for all ranges of $\epsilon$. This suggests that instances arising in practice do have geometric structure favorable to our discrepancy-based BalanceKV algorithm. This holds true for both attention approximation and end-to-end experiments.
>
> 2. When the recursive algorithm (softmaxbalance) is performed, is the error also accumulating exponentially? Eventually the error will depend on the error between the two sets in each partition.
>
> Response: Thank you for your question! The error of MergeAndReduce - the recursive implementation of SoftmaxBalance, depends linearly on the depth of the recursion $T$. We formally prove the theoretical guarantees of MergeAndReduce in Subsection  A.3.2, but briefly this may be explained as follows.
> MergeAndReduce is an online implementation of the following procedure: we split our dataset into blocks of size $t$, compress each block by a factor of 2 with the SoftmaxBalance algorithm and then recurse on the resulting smaller dataset. Please refer to Figure 2 in the paper for visual illustration. The error of transitioning from the $i-1$-st to $i$-th recursion level is upper bounded by the product of $2^i$, the number of blocks of size $t$ into which we partition the dataset at the $i$-th level of recursion, and the error of SoftmaxBalance on one block of size $t$. Note that the number of blocks in the partition of the dataset at the $i$-th level of recursion is upper bounded by $n/(2^i \cdot t)$. Therefore, our upper bound on the error at the $i$-th level of recursion is independent of $i$.
>
> 3. I did not check the appendix, but is the algorithm overall easy to implement? It seems like there will be a lot of extra bookkeeping that might cause more memory usage for larger models.
>
> Response: Thank you for your question! In a nutshell, the algorithm is easy to implement. More precisely, in our experiments, we use BalanceKV to compress the KV cache at the prefill stage. In the streaming part, we generate only a few tokens, therefore  compression is not necessary. As we discuss in more detail in point 1 in the Questions section in the response to Reviewer pBkK, there is a single parameter that we need to tune, and it is quite easy to choose this parameter empirically.

---

> > ### Comment · Reviewer_w4SJ · 2025-08-03
> >
> > I thank the authors for the clarification. I will keep my score as it is.

---

### Official Review · Reviewer_pBkK · 2025-06-30

**Clarity:** 3
**Significance:** 3
**Originality:** 3
**Rating:** 4
**Confidence:** 3

**Summary:**

The paper introduces BalanceKV, a new streaming algorithm for approximating attention in large language models (LLMs) to enable efficient long-context generation. Grounded in discrepancy theory and Banaszczyk’s vector balancing, BalanceKV leverages the geometric properties of key and value tokens to select a balanced subset, achieving $\epsilon$-approximate attention computation with reduced memory overhead. The algorithm provides provable theoretical guarantees on space complexity and approximation error under bounded $l_2$ norm assumptions, alongside a lower bound for streaming attention computation. Empirical evaluations demonstrate BalanceKV’s superior performance over existing methods like SnapKV and PyramidKV on single-layer attention approximation and end-to-end generation tasks across long-context benchmarks, including LongBench and Needle-in-a-Haystack, achieving improved memory efficiency and generation quality.

**Questions:**

1. Theorems 3.1 and 3.4 assume that the $l_2$ norms of query and key vectors are bounded. How are these norms distributed in real LLMs? How does BalanceKV perform if they occasionally exceed the assumed bounds? Can dynamically adjusting the bounds $r$ or layered processing mitigate potential performance degradation?

2. Although the experiments cover context lengths from 16k to 100k tokens, how does BalanceKV perform in multimodal tasks such as ultra-long contexts (e.g., 1M tokens) or video generation? Are there other challenges in these scenarios?

3. In the system efficiency evaluation, BalanceKV has the lowest Prefill Time, but the Decoding Time is slightly higher than StreamingLLM. Can you provide in-depth analysis of the specific reasons for the difference in decoding time? Is there a way to reduce the decoding latency so that it is also optimal in the decoding stage?

**Ethical Concerns:**

["NO or VERY MINOR ethics concerns only"]

**Final Justification:**

based on the strengths and weakness I previously wrote, other reviewers' comments, and  the authors response. I would like to keep  my positive original rating.

**Limitations:**

there is no "limitations" section.

**Quality:**

3

**Strengths And Weaknesses:**

Strengths:

1. This paper applies discrepancy theory and Banaszczyk's vector balance to streaming attention approximation. This geometric balance method has provable approximation guarantees, in contrast to heuristic methods, and establishes a new theoretical benchmark by providing lower bounds on space complexity.

2. This paper provides formal proofs of algorithmic complexity and approximation error bounds (Theorems 3.1 and 3.2), as well as comprehensive empirical evaluations. It consistently outperforms several baselines such as StreamingLLM, SnapKV, and PyramidKV on LongBench, Needle-in-a-Haystack, and single-layer attention tasks.

Weaknesses:

1. The theoretical guarantees (e.g., Theorems 3.1 and 3.4) rely on assumptions like bounded $l_2$ norms for query and key embeddings, which may not always hold in real-world LLM scenarios. The paper lacks discussion or empirical validation on how performance degrades if these assumptions are violated, limiting the understanding of BalanceKV’s robustness across diverse models.

2. The use of a fixed compression rate (0.25) in end-to-end evaluations simplifies experimentation but may not be optimal, as different layers or attention heads might benefit from adaptive compression strategies. Exploring variable compression rates could further enhance performance or memory efficiency but is not addressed.

---

> ### Author Rebuttal · Authors · 2025-07-31
>
> Thank you for carefully studying our submission and the additional, very useful suggestions and comments!
>
> --> The theoretical guarantees (e.g., Theorems 3.1 and 3.4) rely on assumptions like bounded  norms for query and key embeddings, which may not always hold in real-world LLM scenarios. The paper lacks discussion or empirical validation on how performance degrades if these assumptions are violated, limiting the understanding of BalanceKV’s robustness across diverse models.
>
>
> Response: Thanks for pointing this out. Please refer to the 2nd point of response to Reviewer BB75 for the discussion of the upper bounds on the norms of keys and queries in the real data.
> In practice, BalanceKV is robust to the choice of assumed upper bounds on the norms of key and query embeddings. We discuss this more thoroughly in the response to your question 1.
>
> -->The use of a fixed compression rate (0.25) in end-to-end evaluations simplifies experimentation but may not be optimal, as different layers or attention heads might benefit from adaptive compression strategies. Exploring variable compression rates could further enhance performance or memory efficiency but is not addressed.
>
> Response: Thank you for this interesting suggestion! We certainly agree that analysing the properties of different layers, attention heads and designing an adaptive compression scheme based on their importance is an exciting future work direction.
>
>
> Questions:
> 1. Theorems 3.1 and 3.4 assume that the  norms of query and key vectors are bounded. How are these norms distributed in real LLMs? How does BalanceKV perform if they occasionally exceed the assumed bounds? Can dynamically adjusting the bounds  or layered processing mitigate potential performance degradation?
>
> Response: Thank you for the question. For the distribution of norms, please refer  to the 2nd point of response to Reviewer BB75. BalanceKV is empirically very stable. In particular, while norm upper bounds are indeed needed for the theoretical analysis of BalanceKV, its practical performance is stronger. In the practical implementation of our algorithm the expression 1/(2c*R^2), which appears  in line 8, is a single hyperparameter, parameter gamma in our implementation. This is the only hyperparameter that our implementation needs and it is the only parameter which depends on the norms of key and query vectors. We did an ablation study on the parameter gamma, which showed that the range of values of gammas which allows for good empirical results for BalanceKV is quite wide, suggesting strong geometric structure. We will include the ablation study in the final version of the paper. In our experiments, we set gamma to a small constant (about 4).
> Studying adaptive variations of BalanceKV is a great direction; however, as practice suggests, the norms of key and query vectors concentrate well, and imprecise choice of the assumed upper bounds does not lead to performance degradation. Therefore, dynamically updating the assumed upper bound on the norms of embeddings is unlikely to be helpful.
>
>
> 2. Although the experiments cover context lengths from 16k to 100k tokens, how does BalanceKV perform in multimodal tasks such as ultra-long contexts (e.g., 1M tokens) or video generation? Are there other challenges in these scenarios?
>
>
> Response: Thanks for the question. We perform the evaluation of BalanceKV and uniform sampling when applied to the InternVL2.5-8B multimodal LLM for compression rates 1/4  and 1/16, for evaluation on the MS COCO image captioning dataset, the scores are reported below. (The bracket for every method contains the compression rate). The experiment was run on 4 X A100 GPU each with 80 GB VRAM.
> | Method              | Bleu_1 | Bleu_2 | Bleu_3 | Bleu_4 | METEOR | RougeL | CIDEr |
> |---------------------|--------|--------|--------|--------|--------|--------|--------|
> | Exact               | 0.795  | 0.629  | 0.476  | 0.351  | 0.291  | 0.580  | 1.255  |
> | BalanceKV (¼)       | 0.794  | 0.628  | 0.475  | 0.351  | 0.290  | 0.579  | 1.251  |
> | Unif (¼)            | 0.794  | 0.629  | 0.476  | 0.350  | 0.290  | 0.578  | 1.247  |
> | BalanceKV (1/16)    | 0.789  | 0.622  | 0.468  | 0.343  | 0.286  | 0.573  | 1.221  |
> | Unif (1/16)         | 0.789  | 0.619  | 0.465  | 0.340  | 0.284  | 0.571  | 1.207  |
>
> For example, for the CIDEr score (specifically designed for image captioning and evaluates how well a generated caption aligns with a set of human references, higher is better), our method is consistently better than the baseline. The ultra long context or video generation evaluation is also a great suggestion, we will include it in future work.
>
> 3. In the system efficiency evaluation, BalanceKV has the lowest Prefill Time, but the Decoding Time is slightly higher than StreamingLLM. Can you provide in-depth analysis of the specific reasons for the difference in decoding time? Is there a way to reduce the decoding latency so that it is also optimal in the decoding stage?
>
>
> Response: Thank you for the insightful question. Indeed, under idealized conditions, one would expect that decoding time across compression methods to be identical, since the per-token compute in the decoding stage remains the same across methods once KV cache is compressed. However, in practice, several system-level factors can contribute to minor differences in the decoding time, e.g., slicing multi-head KV caches or cache locality. We believe that further system-level tuning (e.g., kernel optimization) could close the gap and potentially make BalanceKV optimal in both prefill and decoding stages.

---

> > ### Author Response · Authors · 2025-08-05
> >
> > Thank you for your supportive evaluation!
> >
> > We hope our responses have addressed your questions and suggestions; please let us know if anything remains unclear.

---

> > ### Comment · Reviewer_pBkK · 2025-08-05
> >
> > Thanks to the authors for the detailed response.

---

### Official Review · Reviewer_BB75 · 2025-07-03

**Clarity:** 3
**Significance:** 3
**Originality:** 3
**Rating:** 5
**Confidence:** 2

**Summary:**

This paper introduces BALANCEKV, a novel streaming algorithm for approximating attention in Transformer-based language models by leveraging tools from discrepancy theory. The key insight is to select a geometrically balanced subset of key-value pairs from the attention cache, thereby reducing memory usage while maintaining high approximation fidelity. The approach uses a variant of the self-balancing walk algorithm to create balanced subsets that preserve softmax-weighted value aggregations. It includes theoretical guarantees for approximation error and memory complexity, and matches known lower bounds up to polylogarithmic factors. The empirical results show consistent improvements over prior methods (e.g., PyramidKV, SnapKV, StreamingLLM) in both approximation accuracy and end-to-end performance across multiple long-context benchmarks including LongBench and Needle-In-A-Haystack.

**Questions:**

N/A

**Ethical Concerns:**

["NO or VERY MINOR ethics concerns only"]

**Final Justification:**

All concerns have been resolved.

**Quality:**

3

**Strengths And Weaknesses:**

### Strengths

1. The paper makes a strong theoretical contribution by connecting streaming attention approximation with vector balancing in discrepancy theory.

2. The algorithm is online and does not require access to future tokens, making it practical for real-time generation.

3. Experiments include comprehensive ablations, runtime vs. error trade-offs, and system-level latency analyses.

4. The method outperforms existing approaches like SnapKV, PyramidKV, and StreamingLLM across various setups and compression rates.

### Weaknesses:

1. The approach performs best when attention weights are relatively smooth; highly peaked attention distributions may suffer more from subsampling, a limitation shared by other pruning methods.

2. Although the paper proves that BALANCEKV is near-optimal in a regime where ε is small and r² ≪ d, it lacks empirical analysis in extremely low-error or very long-context regimes (e.g., >128k tokens).

3. The theoretical guarantees hinge on assumptions that the L2 norms of keys, queries, and values are bounded. Real-world embeddings may violate these, possibly affecting performance.

---

> ### Author Rebuttal · Authors · 2025-07-31
>
> Thank you for carefully studying our submission and the additional, very useful suggestions and comments!
>
> 1. Although the paper proves that BALANCEKV is near-optimal in a regime where ε is small and r² ≪ d, it lacks empirical analysis in extremely low-error or very long-context regimes (e.g., >128k tokens).
>
>
> Response: We think these are great suggestions.  Extremely low error regime corresponds to the low compression rate regime. We conduct the experiment  where we apply BalanceKV and uniform sampling to a few of datasets in LongBench with low compression rates 0.8, 0.9 and 0.95. In our evaluations, we use the model Llama-3.1-8B-Instruct. The experiments are performed on a single A100 GPU with 80GB VRAM. Our experiments show that even in this regime BalanceKV outperforms uniform sampling.
> | Compression Rate |    Dataset     | BalanceKV | Uniform | Baseline |
> |------------------|----------------|-----------|---------|----------|
> | 0.8              | Hotpotqa       | 50.2      | 48.4    | 51.9     |
> | 0.8              | Triviaqa       | 91.6      | 86.3    | 91.6     |
> | 0.9              | Multifieldqa   | 47.5      | 44.9    | 47.8     |
> | 0.9              | Qasper         | 42.3      | 39.6    | 43.1     |
> | 0.95             | Lcc            | 49.3      | 45.7    | 49.5     |
> | 0.95             | P.Count        | 20.7      | 20.1    | 20.7     |
>
> Extremely long context regime is also a very interesting direction, and we will include it in future work.
>
>
> 2. The theoretical guarantees hinge on assumptions that the L2 norms of keys, queries, and values are bounded. Real-world embeddings may violate these, possibly affecting performance.
>
>
> Response: Thank you for raising this excellent point! Indeed, the assumption on the boundedness of the embedding vectors is both important to our theoretical guarantees and stemming from the properties of the real-world datasets.
> In our implementation of BalanceKV, we run the self balancing walk on the datasets of keys shifted by their average. Note that the operation of shifting the key vectors causes both the numerator and the denominator of the softmax function to scale by a constant factor, therefore if our algorithm successfully approximates the attention scores for the shifted keys, then it also performs well for the original set of keys.
> We compute the $\ell_2$ norms of queries and keys shifted by their average, and values (qkv) using the pretrained LLaMA-3.1-8B-Instruct model and Qasper (one of the datasets in LongBench) dataset. Specifically, we analyze each prompt in TriviaQA and compute the average L2 norms of all qkv vectors across all layers and attention heads during the pre-fill stage. Due to the space limit, we provide representative results in the below table from (randomly chosen) prompts of various sequence lengths. The key findings are that all qkv norms consistently concentrate around some constants (~15 for query, and ~15 for key, ~3 for value) with small confidence intervals (CI). Importantly, the norms remain stable across a wide range of sequence lengths, suggesting that these norms do not grow with input length. The table below clearly indicates that the query/key/value norms are well-bounded and stable, validating that our assumption is not only theoretically sound but also holds in practice for real-world language model usage. We will add more detailed results in the final draft.
>
> | Prompt ID | Seq Len | Q Mean  | Q 95% CI | K Shifted Mean  | K 95% CI | V Mean  | V 95% CI |
> |-----------|---------|---------|----------|---------|----------|---------|----------|
> | 10        | 2281    | 14.4512 | 0.0029   | 15.3451 | 0.0065   | 3.3398  | 0.0043   |
> | 24        | 3131    | 14.6003 | 0.0025   | 15.6603 | 0.0058   | 3.3539  | 0.0036   |
> | 30        | 3388    | 14.7406 | 0.0024   | 15.5152 | 0.0054   | 3.3419  | 0.0035   |
> | 22        | 4230    | 14.8339 | 0.0021   | 15.6907 | 0.0049   | 3.3468  | 0.0032   |
> | 5         | 5734    | 14.9259 | 0.0019   | 15.7149 | 0.0041   | 3.3482  | 0.0027   |
> | 14        | 6616    | 14.9000 | 0.0017   | 15.6757 | 0.0038   | 3.3596  | 0.0025   |
> | 4         | 6962    | 14.9743 | 0.0017   | 15.7346 | 0.0039   | 3.3450  | 0.0024   |
> | 21        | 8041    | 14.9364 | 0.0016   | 15.7025 | 0.0035   | 3.3491  | 0.0023   |
> | 26        | 17337   | 15.1437 | 0.0011   | 15.9531 | 0.0024   | 3.3654  | 0.0016   |
> | 27        | 21274   | 15.2065 | 0.0010   | 15.8745 | 0.0022   | 3.3683  | 0.0014   |
>
>
> The intuition is that the qkv embeddings typically can be obtained by multiplying weights by the input embeddings, and those input embeddings are operated through the layer normalization. Therefore, entries of qkv embeddings are expected to be small assuming the weight matrices have small eigenvalues independent of the sequence length. So the norms of qkv embeddings in our case after this normalization are expected to be small constants regardless of the sequence length.

---

> > ### Author Response · Authors · 2025-08-05
> >
> > Thank you for your supportive evaluation!
> >
> > We hope our responses have addressed your questions and suggestions; please let us know if anything remains unclear.

---

> > ### Comment · Reviewer_BB75 · 2025-08-06
> > **Thanks for the response.**
> >
> > Thanks for the response. The authors have resolved my concerns and I will raise my score. Good luck.

---

### Decision · Program_Chairs · 2025-09-17

**Decision:**

Accept (spotlight)

**Comment:**

Although computing attention in transformer-based language models requires $O(nd)$ space to store the keys and values, for context length $n$ and embedding dimension $d$, this paper shows that these bounds can be bypassed when the goal is only to approximate attention, by using tools from discrepancy theory and the streaming model. The algorithm is further complemented with a lower bound and experiments. Initial concerns about the assumption that the $L_2$ norms of keys, values, and queries are bounded were well-addressed by the authors' response.

To strengthen the potential impact of the work, additional action items for the authors include:
- Clarifying the MR-Denominator subroutine, which is currently not described in sufficient detail for independent understanding
- Incorporating the new experiments described in the author rebuttal phase